

# Deep-learning statistical downscaling of precipitation in the middle reaches of the Yellow River: A Residual in Residual Dense Block based network

He Fu[1,2], Jianing Guo[2], Chenguang Deng[2], Heng Liu[2,3], Jie Wu[2], Zhengguo Shi[2,4,5], Cailing Wang[1], and Xiaoning Xie[2,4]

[1]School of Computer Science, Xi'an Shiyou University, Xi'an, 710065, China
[2]SKLLQG, Institute of Earth Environment, Chinese Academy of Sciences, Xi'an, 710061, China
[3]Xi'an Institute for Innovative Earth Environment Research, Xi'an, 710061, China
[4]CAS Center for Excellence in Quaternary Science and Global Change, Xi'an, 710061, China
[5]Institute of Global Environmental Change, Xi'an Jiaotong University, Xi'an, 710049, China

**Correspondence:** Xiaoning Xie (xnxie@ieecas.cn), Cailing Wang (azering@163.com)

**Abstract.** The middle reaches of the Yellow River (MRYR), located in northern China, are the transition zone between semi-arid and semi-humid climates. As one of the climate-sensitive regions in China, MRYR has fragile ecological environment and serious soil loss, which leads to geological disasters such as landslides, collapses and mudslides caused by extreme precipitation to occur. However, scarceness of the high-resolution precipitation data over MRYR limits the assessment of the environmental impacts caused by climate change, especially for extreme precipitation. In this paper, we design a Residual in Residual Dense Block based network (RRDBNet) model for the statistical downscaling of precipitation in MRYR, and compare the proposed RRDBNet with the generalized linear regression model and two popular deep learning-based models. The results show that the proposed RRDBNet model has a good performance on precipitation simulations, which can well reproduce the spatial-temporal characteristics of high-resolution precipitation. Especially, RRDBNet has substantial improvements in extreme precipitation compared with other models. On the probability density function of daily precipitation, it is further demonstrated that RRDBNet performs better on extreme precipitation frequency. Our results suggest that the statistical downscaling based on RRDBNet may be an effective tool for historical and future climate simulations from global climate models.

## 1 Introduction

The middle reaches of the Yellow River (MRYR) are located in the transition zone between semi-arid and semi-humid climates, where the precipitation gradually increases from northwestern to southeastern regions. In MRYR, the precipitation mainly concentrates in summer, followed by autumn. Due to fragile ecological environment and serious soil loss, geological disasters, such as landslides, collapses and mudslides, are always linked to heavy precipitation events during summer (Li et al., 2010; Zhuang et al., 2017). In response to global warming, extreme weather and climate events become more and more frequent (Prein et al., 2017; Zhang et al., 2013), which may increase the possibilities of major geological hazards in the region. To





further assess the environmental impacts caused by climate change (especially for extreme precipitation), it is crucial to obtain the accurate, reliable and high-resolution precipitation data over the MRYR region.

Due to the coarse spatial resolutions (usually ∼100 km) and oversimplified parameterizations in Global Climate Models (GCMs) (Berg et al., 2013), downscaling techniques have been widely applied for GCM outputs to meet the demands of subgrid-scale details of regional precipitation. Downscaling techniques can be broadly classified into two categories, namely,

dynamic downscaling and statistical downscaling (Fowler et al., 2007). Dynamic downscaling methods are physical-based and designed to transform coarse grid outputs from GCMs into higher-resolution data using regional climate models (RCMs) (Schmidli et al., 2006). In dynamic downscaling, RCMs are nested within GCMs or utilize the output of GCMs as a boundary condition for further simulations to generate higher-resolution climate. RCMs with different resolutions and parameterization schemes have been applied in China to assess the recent regional climate change and future projections (Bao et al., 2015;

Ma et al., 2015; Tian et al., 2020; Tang et al., 2023). However, accurate precipitation simulations have remained a challenge since the ability of RCMs highly relies on the optimal combination of physical parameterizations (Efstathiou et al., 2013). The inaccurate boundary condition from GCMs also results in biased extreme precipitation (Tian et al., 2020). More important, RCMs are expensive and time-consuming to compute, which usually require high performance computing resources (Benestad and Haugen, 2007; Benestad, 2010; Kaur et al., 2020).

Statistical downscaling can be easily applied to dealing with the output data from GCMs, which employs fewer computational processes and consumes less computational resources (Chen et al., 2012). Xu and Wang (2019) used a quantile mapping method to bias correct and downscale the Chinese daily maximum temperature for 13 GCMs in the Coupled Model Intercomparison Project Phase 5 (CMIP5), and found that the method significantly improves the original CMIP5 climate projections. Panjwani et al. (2021) compared two methods of quantile mapping techniques including the linear and the tricube methods,

suggesting that the tricube method performed better in removing bias from climate data. Furthermore, many regression models (Xu, 1999) such as multiple linear regression (Gutiérrez et al., 2013), and generalized linear models (San-Martín et al., 2017) are developed based on the statistical relationship between regional-scale meteorological predictors and circulation characteristics. Although the traditional methods are easy to understand and interpret, they cannot accurately characterize the nonlinear dependence among climate variables due to their simple assumptions (Booij, 2002; Beniston et al., 2007; Prudhomme and

Davies, 2009).

Currently, data-driven methods based on deep learning have been tried in statistical downscaling since they are skilled in handling nonlinear relationships and have adaptive and generalized capabilities. The data-driven models have performed well in downscaling experiments on different meteorological variables related to temperature and precipitation (Sachindra et al., 2018; Li et al., 2020; Anaraki et al., 2021). Vandal et al. (2017) used a stacked super-resolution convolutional neural network

(CNN) to construct a framework DeepSD for statistical downscaling precipitation. To improve the quality of forecasts, many models about CNNs are employed to generate high-resolution data (Rodrigues et al., 2018; Pan et al., 2019; Baño-Medina et al., 2020). It was demonstrated that CNNs have better performance on the estimation of precipitation than reanalysis products and statistical downscaling products obtained using linear regression, nearest neighbor and random forest (Pan et al., 2019; Baño-Medina et al., 2020). In addition, there are several applications of deep learning in other areas of meteorology, such as tropical





cyclone intensity estimation (Pradhan et al., 2017), real-time forecasting (McGovern et al., 2017) and physical parameterization (Brenowitz and Bretherton, 2019).

In this paper, we develop a deep learning-based statistical downscaling method, i.e., Residual in Residual Dense Block based network (RRDBNet) model, to produce daily high-resolution precipitation over the MRYR region, and the performance of RRDBNet is compared with the generalized linear regression model (GLM) and other two deep learning-based models. The

paper is organized as follows: section 2 introduces the study area and dataset; the proposed modeling approach, experimental setup and evaluation metrics are described in detail in section 3; the results are illustrated in section 4. Finally, the conclusions are given in section 5.

## 2   Study area and data

In this paper, the study area, i.e., MRYR, is defined as ($32°N - 42°N$, $100°E - 116°E$ ) (Fig. 1). The input dataset (predictor

set) of statistic downscaling is derived from ERA5. ERA5 is the latest generation of atmospheric reanalysis data published by the European Centre for Medium-Range Weather Forecasts (ECMWF). Compared with previous generations from ECMWF, ERA5 has better improvements in model physics, core dynamics, data assimilation techniques, and the increase in data sources, which can more accurately reproduce atmospheric conditions (Hersbach et al., 2020). The predictors (input variables) include daily horizontal wind components, geopotential height, specific humidity, and temperature, which has a $2° \times 2°$ horizontal

resolution at three pressure levels (850, 700, and 500 hPa). The five predictors are stacked together to constitute the input data matrix $X$ in Fig. 2.

The daily precipitation is derived from the latest Integrated Multi-satellite Retrievals for Global Precipitation Measurement (IMERG) half-hourly precipitation data (http://gpm.nasa.gov/data-access/downloads/gpm) (Huffman et al., 2015). The half-hourly IMERG precipitation data with a spatial resolution of $0.1°$ (about 10 km) is accumulated every day (UTC 00:00 to UTC

23:30), chosen as the target prediction $Y$. The algorithm for IMERG is developed by Huffman et al. (2019), which combines retrievals from multi-satellite Global Precipitation Measurements (GPM) product and ground-based instruments to provide high-quality precipitation estimates. The GPM precipitation dataset calibrated with monthly gauge-based analysis data, and showed excellent performance in detecting heavy rain events in China (Su et al., 2018). The time period of the study is from 2001 to 2020. The data from 2001 to 2015 was used for training models and the data from 2016 to 2020 was employed to

test models. Observed daily precipitation data of 149 in-situ stations in MRYR (red dots in Fig. 1b) covering Jan 1st, 2016 to Dec 31th, 2019 (obtained from the China Meteorological Administration, https://data.cma.nc) is also applied to validate the IMERG precipitation and test models.



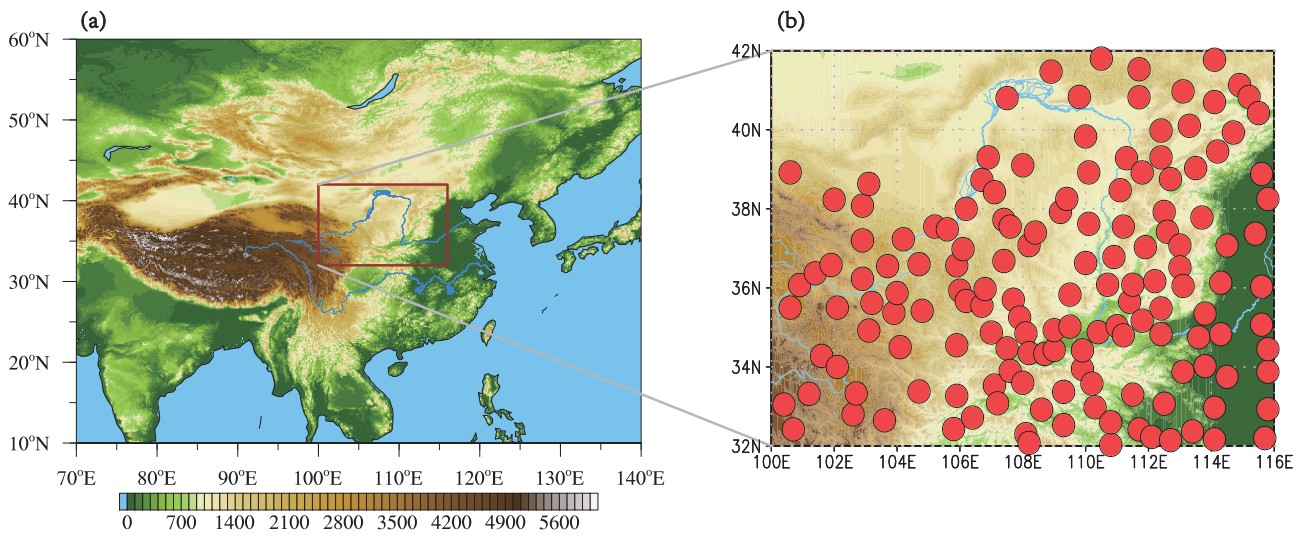

**Figure 1.** (a) The study area (red solid line rectangles) and the topographic features (unit: m) in the middle reaches of Yellow River. (b) The location of 149 in-situ stations (red dots).

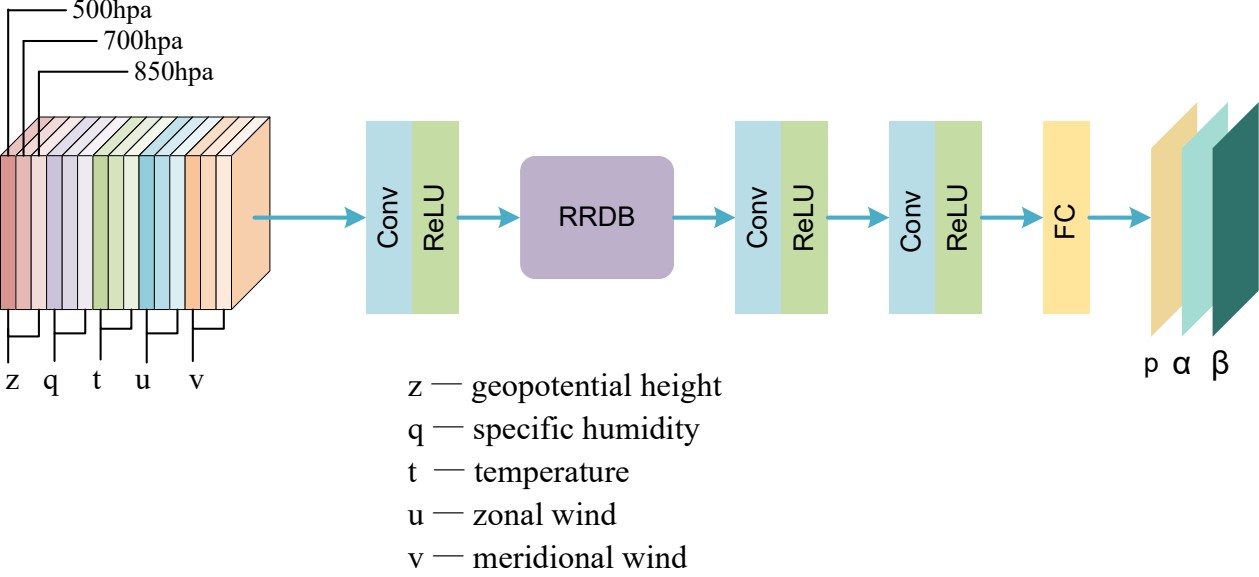

**Figure 2.** RRDBNet overall framework flow chart.





## 3 Methodology

### 3.1 RRDBNet Model

Fig. 2 shows the general structure of RRDBNet, in which the predictor set $X \in \mathbb{R}^{6 \times 9 \times 15}$ needs to be processed first by a convolution (Conv) and activation function ReLU (Nair and Hinton, 2010) to extract shallow features. Then, the Residual in Residual Dense Block (RRDB) (Wang et al., 2018) is used to extract rich local features and adaptively learn more effective feature information from the previous and current fused features. Finally, two consecutive Conv, ReLU and a fully connected layer (FC) are used to obtain parameters $p$, $\alpha$ and $\beta$. The final precipitation $Y \in \mathbb{R}^{t \times 100 \times 159}$ is obtained by calculating the

parameters $p$, $\alpha$ and $\beta$.

#### 3.1.1 Network Architecture

The network model consists of three parts: (1) Shallow feature extraction network, which consists of a Conv and ReLU. (2) RRDB, which contains three Residual Dense Block (RDB) and one residual edge, each RDB in turn contains five Conv and four LReLU (Maas et al., 2013). (3) To obtain the parameters $p$, $\alpha$ and $\beta$, this part consists of two consecutive Conv, ReLU and

a FC. Table 1 shows the parameters related to the convolution kernel in each convolutional layer. The convolution operations used in this paper are all two-dimensional convolutions.

**Table 1.** Convolutional kernel shape and size in each convolutional layer.

| Convolution kernel | Kernels | Kernel shapes | Stride shapes |
|---|---|---|---|
| | | $k_1^i \times k_2^i$ | $s_1^i \times s_2^i$ |
| Conv2D | 32 | $(3,3)$ | $(1,1)$ |
| Each Conv2D in RRDB | 32 | $(3,3)$ | $(1,1)$ |
| Conv2D | 25 | $(3,3)$ | $(1,1)$ |
| Conv2D | 3 | $(3,3)$ | $(1,1)$ |

The purpose of convolution is to perform feature extraction and filter out unwanted information. Two-dimensional convolution is used to extract the rich and abstract spatial feature information between predictors. The convolution operation is specifically a dot-product operation of the input data and the convolution kernel, which is shifted in certain steps to cover

the entire spatial dimension of the input data. The nonlinearity is introduced into the model by applying the corresponding activation function to the feature information obtained from the convolution. The specific process can be expressed as follows:

$$v_{i,j}^{x,y} = ReLU\left(b_{i,j} + \sum_{\tau=1}^{d_{l-1}} \sum_{\rho=-\gamma}^{\gamma} \sum_{\sigma=-\delta}^{\delta} \omega_{i,j,\tau}^{\sigma,\rho} \times v_{i-1,\tau}^{x+\sigma,y+\rho}\right), \tag{1}$$



where $v_{i,j}^{x,y}$ represents the activation value with spatial location coordinates $(x, y)$ in the $j^{th}$ feature map of the $i^{th}$ layer, $d_{l-1}$ represents the number of feature maps in the $(i-1)^{th}$ layer, $2\delta + 1$ and $2\gamma + 1$ represent the height and width of the convolution kernel, respectively. $\omega_{i,j}$ is the convolution kernel of the $j^{th}$ feature map of the $i^{th}$ layer. $ReLU$ is the activation function from Nair and Hinton (2010). The role of $ReLU$ is to introduce nonlinear factors into the model to improve the expressive ability of the network model.

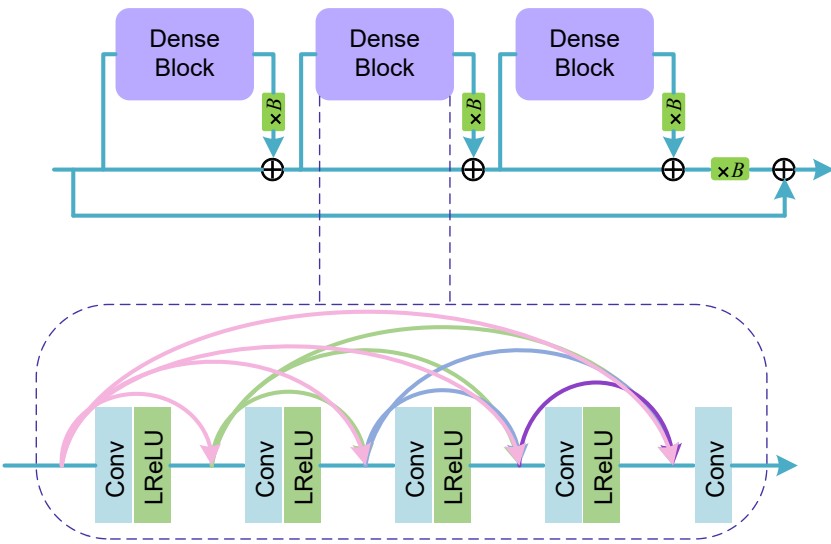

**Figure 3.** Detailed structure of Residual in Residual Dense Block (RRDB).

Fig. 3 shows the detailed structure of RRDB, which employs a multi-level residual learning and dense connectivity strategy. It is composed of a backbone network and a residual edge. The output information flow of the backbone network is superimposed with the information flow in the residual edge and transmitted backward. The backbone network contains three RDBs, each of which is equivalent to the combination of Residual Block and Dense Block. Dense concatenation between convolutional layers in the RDB is performed to extract rich local feature information. The states learned by the previous RDB are concatenated with all layers of the current RDB, and then more effective information is learned adaptively from the previous and current local fusion features. Furthermore, the Batch Normalization (BN) (Ioffe and Szegedy, 2015) layer is not included in RRDB. During training, BN layer is used to batch normalize the data and save the feature means and variances. When the mean and variance obtained during training are used to normalize the test set during testing, the BN layer introduces artifacts that limit the generalization ability of the network if the statistics of the test and training sets are very different. Removing the BN layer enhances the expressiveness of the network, reduces the computational complexity, and saves computational resources and memory usage to some extent. Residual scaling (Lim et al., 2017; Szegedy et al., 2017) was also added to RRDB. Residual scaling means that the learned residuals are multiplied by a constant between 0 and 1 and then added to the feature information on the main path, which has been used to enhance the stability of the network. As shown in Fig. 3, in this paper we set $B = 0.2$. The activation function used in RRDB is LReLU.



### 3.1.2 Bernoulli-gamma distribution

A probabilistic modeling is the current downscaling framework of precipitation (Cannon, 2008), instead of deterministic mod-
eling (Bürger, 1996, 2002). Deterministic predictions often underestimate the extreme precipitation (Baño-Medina et al., 2020;
Cannon, 2008; Williams, 1997; Schoof and Pryor, 2001; Maraun and Widmann, 2018). Therefore, we study stochastic precip-
itation prediction in this paper. Due to the mixed discrete and continuous nature of precipitation, Williams (1997) suggested
using the bernoulli-gamma distribution to describe precipitation, which has been used for single-site precipitation downscaling
models (Haylock et al., 2006; Cawley et al., 2007). The probability density function of the Bernoulli-gamma distribution is
expressed as

$$f\left(y;p,\alpha,\beta\right)=\begin{cases}1-p & \text{for } y=0 \\ \frac{py^{\alpha-1}exp(-y/\beta)}{\beta^{\alpha}\Gamma(\alpha)} & \text{for } y>0\end{cases}, \tag{2}$$

where $y$ represents the observed precipitation, $\alpha\left(\alpha>0\right)$ and $\beta\left(\beta>0\right)$ represent the shape and scale of the gamma distribution
function, respectively. $p$ is the probability of rain (Bernoulli distribution parameter) and $\Gamma\left(\cdot\right)$ represents the gamma function.

After the FC layer, the output of the grid points for each target prediction can be expressed as follows:

$$p_i\left(t\right)=o_i^1\left(t\right), \tag{3}$$

$$\alpha_i\left(t\right)=exp\left[o_i^2\left(t\right)\right], \tag{4}$$

$$\beta_i\left(t\right)=exp\left[o_i^3\left(t\right)\right], \tag{5}$$

where $o_i^1\left(t\right)$, $o_i^2\left(t\right)$ and $o_i^3\left(t\right)$ represent the three output values of a target grid point $i$ at time $t$, respectively. The final precipi-
tation is obtained by multiplying $p$ with the random values of the distribution with shape $\alpha$ scale $\beta$.

### 3.1.3 Loss function

The loss function is an arithmetic function that measures the difference between the model's predicted value and the true value.
In the training phase, In the training phase, when the model outputs predicted values by forward propagation, the loss function
calculates the difference between the predicted and actual values, i.e., the loss value. After obtaining the loss value, the model
updates the parameters of the network by back propagation to reduce the loss between the predicted and true values, thus
guiding the training in the right direction for the purpose of learning. The loss function $L$ (Cannon, 2008; Williams, 1997;
Haylock et al., 2006; Cawley et al., 2007) adopted in this study is as follows:

$$L=-\sum_{t=1}^{N}\sum_{m=1}^{M}log\left\{f_m\left[\left(y_m\left(t\right)|x\left(t\right)\right)\right]\right\}, \tag{6}$$

where $x\left(t\right)$ represents the set of predictors input to the model at time $t$, and $y_m\left(t\right)$ is the observed precipitation at grid point
$m$ at time $t$. $f_m\left(\cdot\right)$ represents the probability density function of the Bernoulli-gamma distribution, $M$ is the total number of
target grid points, and $N$ is the batch time step.



The purpose of training is to minimize the negative log likelihood of the loss function $L$ and update the network model parameters by back propagation. The training process of the whole model is shown in Fig. 4. $I$ represents the number of predictors and $M$ is the total number of target grid points in Fig. 4.

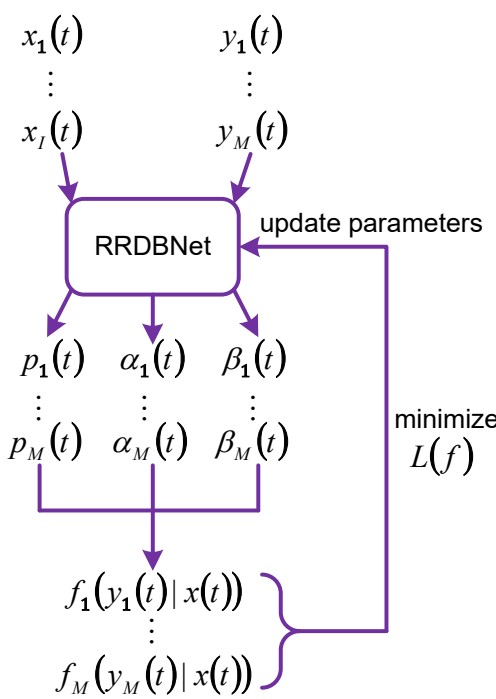

**Figure 4.** Flow chart of the training process.

### 3.1.4 Experimental parameter setting

155   The network structure proposed in this experiment is shown in Fig. 2, where the specific parameters of each layer are configured as displayed in Table 1. In this paper, we use the Adam (Kingma and Ba, 2014) optimizer to optimally update the weights of the convolution filter, the learning rate set to $10^{-4}$, and the batch size set to 32. The loss function uses the function defined by Eq. 6 to minimize the negative log-likelihood of the Bernoulli-gamma distribution. We use the early stopping strategy to regularize the model to prevent overfitting, where patience is set to 30. During the training period, the best model weight parameters are

160   saved. And during testing period, the model calls this best parameter weight to test the data.

### 3.2 Models for comparison

We compare the proposed RRDBNet with a GLM (Baño-Medina et al., 2020; Bedia et al., 2020) method and two deep learning-based methods including CNN (Baño-Medina et al., 2020), and RDBNet (Zhang et al., 2018; Wang et al., 2018).





**GLM**: Generalized linear regression model(GLM) uses binomial family and linked logit to predict the occurrence of
precipitation and gamma-based family and linked logit to predict the amount of rainfall, respectively. Finally, the occurrence
of rainfall is multiplied by the amount of rainfall to obtain the final rainfall forecast. In this paper, the GLM considers the
predictors of the four grids that are nearest neighbors to the target location (Baño-Medina et al., 2020).

**CNN**: The CNN is derived from the model CNN1 proposed by Baño-Medina et al. (2020) for statistical downscaling of
precipitation prediction. Baño-Medina et al. (2020) also compared the CNN1 with some variants of the CNN model and found
that the CNN1 performs well in the European domain for precipitation downscaling.

**RDBNet**: The RRDB in the proposed RRDBNet is composed of RDB performing multilevel residual learning. We replace
the RRDB with a single RDB, and the network model formed after the replacement is denoted as RDBNet. RDBNet is em-
ployed as a comparison model to verify whether the multi-level residual learning strategy of RRDB is effective in precipitation
downscaling.

## 3.3 Evaluation matrix

To comprehensively evaluate the performance of each method, we used three evaluation metrics: Difference (unit: mm/day),
root mean squared error (RMSE) (unit: mm/day), and the Pearson correlation coefficient (CC). The detailed equations are as
follows:

$$Difference = \frac{1}{n}\sum_{i=1}^{n} X_P - X_G, \tag{7}$$

$$RMSE = \sqrt{\frac{1}{n}\sum_{i=1}^{n}(X_P - X_G)^2}, \tag{8}$$

$$CC = \frac{\sum_{i=1}^{n}(X_P - \bar{X_P})(X_G - \bar{X_G})}{\sqrt{\sum_{i=1}^{n}(X_P - \bar{X_P})^2}\sqrt{\sum_{i=1}^{n}(X_G - \bar{X_G})^2}}, \tag{9}$$

Where $X_P$ represents the data predicted by the model and $X_G$ is the GPM data. $n$ is the total numbers of grid points divided
in the area. $\bar{X_P}$ and $\bar{X_G}$ are the mean value of the model prediction data and GPM data, respectively.

## 4 Results

## 4.1 Precipitation

To examine the performance of different models, annual and seasonal precipitation, as well as temporal variation of monthly
precipitation are evaluated against observations. Fig. 5 shows the spatial distribution of annual mean precipitation for GPM and
downscaling projections based on the four models from 2016 and 2020. It is shown that less precipitation in the northwestern
region while more rainfall in the southeastern region in Fig. 5a. The spatial pattern shows a gradual increase in precipitation
from the northwest to southeast regions. Meanwhile, it is clearly shown that the spatial pattern of annual mean precipitation
for GLM, CNN, RDBNet and RRDBNet are all similar to that of GPM. All four models can mainly capture this pattern of
gradually increasing precipitation from northwest to southeast regions. The percentage differences in these four models are





shown in Fig. 6. GLM, CNN, and RDBNet show a significant overestimation in the central and northwestern regions of the MRYR, and the proposed RRDBNet has a significant improvement in these regions compared to other models. Table 2 shows

the evaluation metrics of annual precipitation for these four models. Three deep learning-based models are closer to GPM data than GLM (-0.12 mm/day) on difference. The proposed RRDBNet outperforms the other models in both RMSE and CC, achieving the smallest RMSE (0.29 mm/day) and the largest CC (0.94), respectively. The smallest RMSE and largest CC indicate that RRDBNet has higher agreement with GPM precipitation. In a word, RRDBNet shows an improvement in the spatial characteristics of annual precipitation in the MRYR compared with other models.

Seasonal variations of precipitation are shown in Fig. 7. It can be seen that precipitation is concentrated in summer and autumn, and less in spring and winter in GPM, which is also displayed in the four models (Fig. 7). We further calculated the related metrics about precipitation in summer and autumn, as shown in Table 3. RRDBNet was all optimal in Difference (-0.02 mm/day), RMSE (0.70 mm/day) and CC (0.89), suggesting an improvement compared to GLM, CNN and RDBNet for summer precipitation. In the autumn, Table 3 shows that the deep learning-based model has a large improvement over GLM in terms of Difference, RMSE and CC, with RRDBNet showing the best performance among them.

**Table 2.** Evaluation metrics of annual mean precipitation in different models.

| Models | Difference | RMSE | CC |
|--------|-----------|------|-----|
| GLM | -0.12 | 0.36 | 0.89 |
| CNN | 0.07 | 0.35 | 0.92 |
| RDBNet | 0.03 | 0.33 | 0.90 |
| RRDBNet | -0.10 | 0.29 | 0.94 |

**Table 3.** Evaluation metrics of summer and autumn precipitation in different models.

| Models | Summer (JJA) | | | Autumn (SON) | | |
|--------|------------|------|-----|------------|------|-----|
| | Difference | RMSE | CC | Difference | RMSE | CC |
| GLM | 0.05 | 0.74 | 0.85 | -0.07 | 0.86 | 0.67 |
| CNN | 0.39 | 0.92 | 0.86 | 0.34 | 0.69 | 0.90 |
| RDBNet | 0.37 | 0.99 | 0.78 | 0.27 | 0.55 | 0.89 |
| RRDBNet | -0.02 | 0.70 | 0.89 | 0.14 | 0.44 | 0.90 |






**Figure 5.** Spatial distribution of annual mean precipitation (mm/day) in the MRYR from 2016 to 2020 for (a) GPM, (b) GLM, (c) CNN, (d) RDBNet, and (e) RRDBNet.



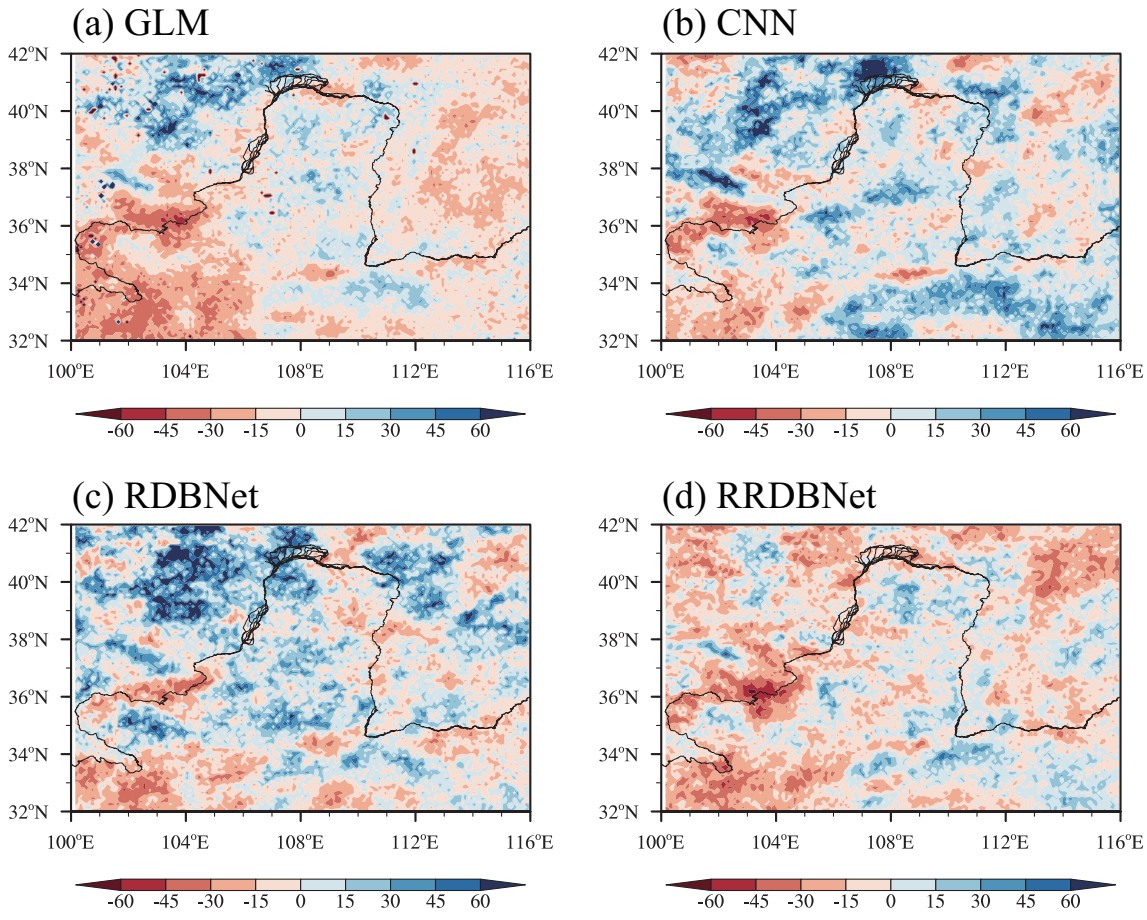

**Figure 6.** Percentage differences in annual precipitation (%) between model and GPM from 2016 to 2020 in the MRYR for (a) GLM, (b) CNN, (c) RDBNet, and (d) RRDBNet. $Percentage\ Difference = (MODEL - GPM)/GPM \times 100\%$.

**Table 4.** Evaluation metrics of variations of monthly averaged precipitation from 2016 to 2020 in different models.

| Models | Difference | RMSE | CC |
|---|---|---|---|
| GLM | -0.12 | 0.55 | 0.91 |
| CNN | 0.07 | 0.47 | 0.96 |
| RDBNet | 0.03 | 0.52 | 0.94 |
| RRDBNet | -0.10 | 0.38 | 0.96 |

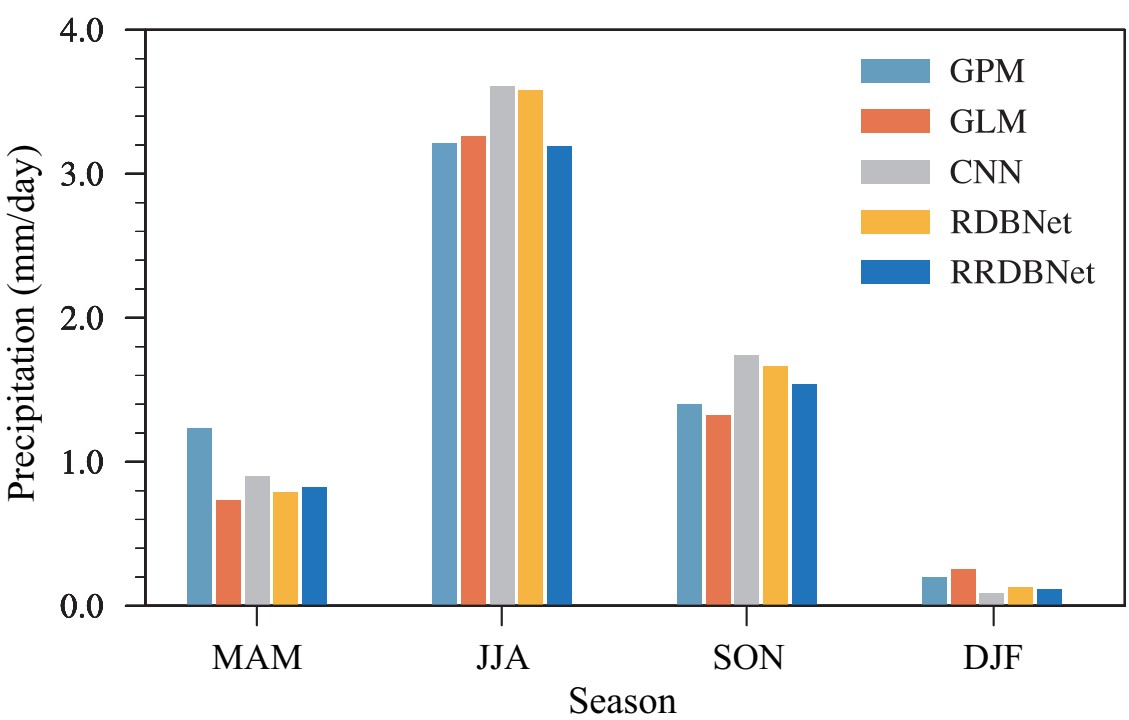

**Figure 7.** Seasonal variation of precipitation (mm/day) in the MRYR from 2016 to 2020 for GPM, GLM, CNN, RDBNet, and RRDBNet.

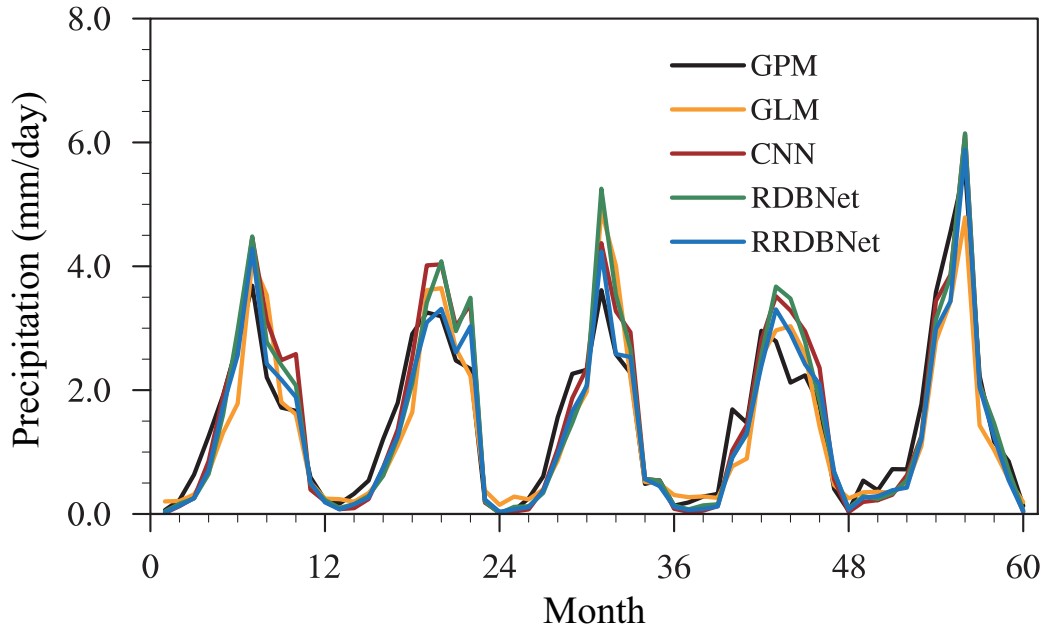

**Figure 8.** Variations of monthly mean precipitation (mm/day) in the MRYR during 2016-2020 for GPM, GLM, CNN, RDBNet, and RRDB-Net.





In addition, we assess the temporal variations of monthly average precipitation from 2016 to 2020 in Fig. 8. The GLM, CNN, RDBNet, and RRDBNet mainly capture the monthly variations of precipitation in the MRYR. To show the performance of the models, we calculated the evaluation metrics of each model on the monthly scale as shown in Table 4. It is found that all three deep learning-based methods outperform the GLM in evaluation metrics on monthly scale, indicating that the

deep learning-based models have strong competitiveness in precipitation estimation. It is noted that the RRDBNet outperforms other models in both RMSE and CC. The smallest RMSE (0.38 mm/day) and the largest CC (0.96) indicate that the RRDBNet performs better in reproducing monthly precipitation variability, suggesting a substantial improvement compared to other models. In summary, it can be concluded from the analysis in annual, seasonal precipitation and temporal variation of monthly precipitation that RRDBNet excels in capturing the variation of spatial and temporal characteristics of precipitation in the

MRYR, and have improvements compared with other models.

### 4.2   Probability distribution function of daily precipitation

To comprehensively evaluate the performances of each model in precipitation variations, we further investigated the probability density function (PDF) of precipitation in the MRYR. As shown in Fig. 9a, CNN, RDBNet and RRDBNet all showed underestimation for daily precipitation below 3 mm, and followed by overestimation from 3-25 mm, while GLM showed the opposite

phenomenon. For the range from 25 mm-50 mm, RRDBNet and GLM are very close to GPM. In the heavy precipitation and extreme precipitation (50-100 mm), GLM, CNN and RDBNet all have an obvious overestimation phenomenon in Fig. 9b. The RRDBNet is closer to the GPM data, with a good fit and higher consistency. In addition, RRDBNet is also the closest to GPM in the precipitation frequency above 100 mm. Based on PDF of daily precipitation, the proposed RRDBNet performs well in heavy and extreme precipitation. Furthermore, we also compare the PDF of precipitation in the MRYR derived from GPM and

RRDBNet with observations at the station sites in Fig.1. It shows that the PDF of GPM precipitation is well consistent with station observations in Fig. 9c and Fig. 9d. These figures indicate that the RRDBNet model can mainly capture the PDF of observations.

### 4.3   Extreme precipitation

To evaluate the performance of each model in extreme precipitation, we selected two climate indices of defined by extreme

precipitation i.e., R95P and R99P. R95P represents the annual cumulative precipitation with daily precipitation greater than the 95th percentile of precipitation on wet days (larger than 1 mm). R99P is the annual cumulative precipitation with daily precipitation greater than the 99th percentile of precipitation on wet days.

Fig. 10 shows the spatial pattern of R95P in GPM and models. In Fig. 10(a), the values of R95P are smaller in the northwestern region of the MRYR and larger in the southeastern region. It shows a gradual increase from the northwest to southeast

regions in R95P and all four models have similar spatial pattern. We show the percentage differences in R95P between the models and GPM in Fig. 11. It is clearly shown that the GLM shows an obvious overestimation in the central region of the MRYR, while CNN, RDBNet, and RRDBNet all showed significant improvements compared to the GLM on this region. At the same time, the performance of RRDBNet in capturing extreme precipitation is optimal among three deep learning-based



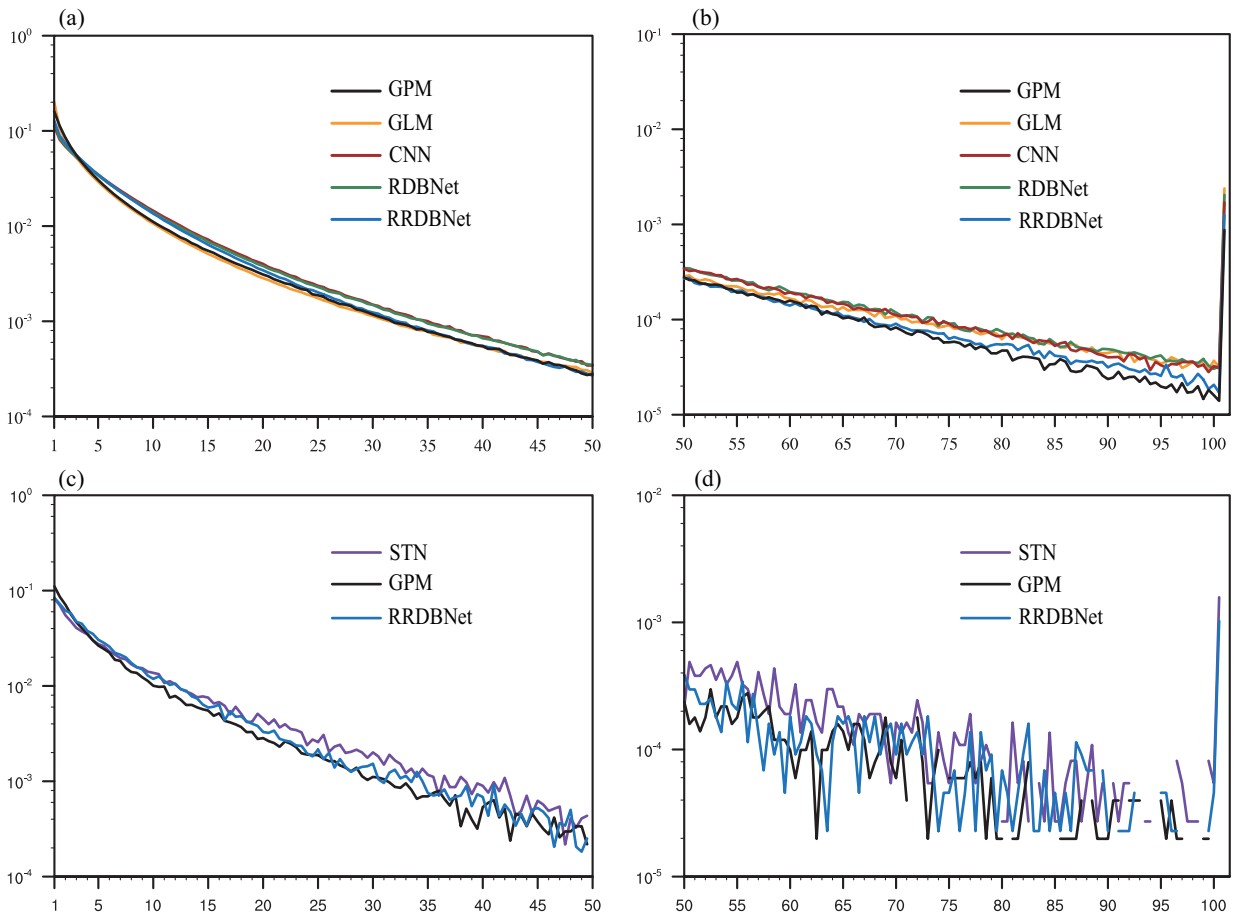

**Figure 9.** (a) Comparison of probability density functions of daily precipitation from 1 mm to 50 mm for all grids of GPM, GLM, CNN, RDBNet, and RRDBNet in the MRYR from 2016 to 2020. (b) The same as (a) but from 50 mm to 100 mm. (c) and (d) The same as (a) and (b) but for observation from 149 stations and nearest grids of GPM and RRDBNet in the MRYR from 2016 to 2019.

models, with the most significant improvement in the central region. We further compared the Difference, RMSE and CC

between each model and GPM as shown in Table 5. It is obvious that three deep learning-based models are far superior to the
GLM on Difference, RMSE, and CC. Note that RRDBNet has the smallest RMSE (39.06 mm/day) and the largest CC (0.90) in
R95P, suggesting that best simulation of R95P in this model. As shown in Fig. 12, all models mainly capture the spatial pattern
of R99P. The percentage differences in R99P between models and the GPM are shown in Fig. 13. Table 6 shows the RMSE
and CC of R99P values between models and the GPM. RRDBNet (RMSE=18.02 mm/day; CC=0.81) is not only significantly

improved compared to GLM (RMSE=85.73 mm/day; CC=0.33), but also optimal compared to CNN (RMSE=18.74 mm/day;
CC=0.81), RDBNet (RMSE=23.67 mm/day; CC=0.75). Totally, it can be concluded that RRDBNet performs optimally in the
spatial distribution characteristics of extreme precipitation R95P and R99P.



**Figure 10.** Spatial distribution of R95P in (mm) the MRYR from 2016 to 2020 for (a) GPM, (b) GLM, (c) CNN, (d) RDBNet, and (e) RRDBNet.



**Figure 11.** Percentage differences in R95P (%) between model and GPM from 2016 to 2020 in the MRYR for (a) GLM, (b) CNN, (c) RDBNet, and (d) RRDBNet.

**Table 5.** Comparison of model evaluation metrics in R95P.

| Models | Difference | RMSE | CC |
|--------|-----------|------|-----|
| GLM | 16.41 | 94.08 | 0.65 |
| CNN | 1.74 | 41.46 | 0.90 |
| RDBNet | 5.91 | 45.85 | 0.86 |
| RRDBNet | -12.86 | 39.06 | 0.90 |





**Figure 12.** Spatial distribution of R99P (mm) in the MRYR from 2016 to 2020 for (a) GPM, (b) GLM, (c) CNN, (d) RDBNet, and (e) RRDBNet.

**Figure 13.** Percentage differences in R99P (%) between model and GPM from 2016 to 2020 in the MRYR for (a) GLM, (b) CNN, (c) RDBNet, and (d) RRDBNet.

**Table 6.** Comparison of model evaluation metrics in R99P.

| Models | Difference | RMSE | CC |
|---|---|---|---|
| GLM | 13.16 | 85.73 | 0.33 |
| CNN | 0.31 | 18.74 | 0.81 |
| RDBNet | 3.37 | 23.67 | 0.75 |
| RRDBNet | -3.38 | 18.02 | 0.81 |





We further show seasonal variations of extreme precipitation as shown in Fig. 14. It is shown that the extreme precipitation including R95P and R99P is concentrated in summer and autumn, and less in spring and winter. We calculated the evaluation

metrics of Difference, RMSE, and CC of R95P and R99P in summer and autumn, as shown in Table 7. In the summer, R95P (RMSE=36.65 mm/day; CC=0.80) and R99P (RMSE=20.42 mm/day; CC=0.65) of RRDBNet are all optimal in RMSE and CC. In addition, R95P and R99P of the deep learning-based model have significant improvements in both Difference and RMSE compared to the GLM. R95P and R99P in RRDBNet have significant improvements in Difference, RMSE and CC compared to GLM. In the autumn, Table 7 shows the R95P and R99P of the deep learning-based models with large improvements in both

RMSE and CC. The RMSE values of RRDBNet for R95P (RMSE=22.90 mm/day) and R99P (RMSE=13.42 mm/day) are the smallest.

Fig. 15 shows the variations of monthly mean extreme precipitation of the models and observations. It can be seen that each model can capture the temporal variations of extreme precipitation in the MRYR. To evaluate the performances of the models, we calculated the evaluation metrics of each model on the monthly-scale extreme precipitation as shown in Table 8.

The smallest RMSE (R95P: 7.12 mm/day; R99P: 3.18 mm/day) and the largest CC (R95P: 0.91; R99P: 0.86) indicate that the RRDBNet outperforms the other models and performs well in reproducing the changes in extreme precipitation. In extreme precipitation, RRDBNet is more consistent with the GPM data and has a substantial improvement compared to other models, and can capture temporal characteristics of extreme precipitation.

The spatial characteristics of precipitation in the MRYR show an increase from the northwestern to southeastern regions.

To further evaluate the model's performance in simulating precipitation in various regions within the MRYR, we divided the MRYR into four zones equally along the midline, i.e., northwest, southwest, northeast, and southeast zones. The results show that RRDBNet can well simulate the precipitation frequency under different precipitation intensities in each zone (figures not shown), which is proved to successfully derive the optimal PDF of daily precipitation in different zones.

By systematically evaluating the R95P and R99P metrics, we can infer that RRDBNet is not only greatly improved com-

pared to the GLM in the spatial and temporal characteristics of extreme precipitation in the MRYR region, but is also optimal compared to CNN, and RDBNet. It shows that RRDBNet is very effective in extreme precipitation simulations, accurately reproducing extreme precipitation events and successfully capturing most of the atmospheric circulation patterns related to extreme precipitation. Thus, RRDBNet is very effective in precipitation downscaling simulations, which can accurately reproduce the spatial and temporal variability of precipitation, especially for extreme precipitation.

**4.4  Comparison of convergence**

From the above subsection, it can be concluded that among the deep learning-based models, RRDBNet is optimal for precipitation downscaling. CNN is superior to RDBNet in most cases. In this subsection, we further conduct comparative experiments on the model convergence of RRDBNet and CNN during training. To ensure the fairness of the comparison, each model is running on the same CPU server, both using the early stopping policy, where patience is set to 30. Other parameters are set as

described in subsection 3.1.4. The result is shown in Fig. 16. It can be seen that RRDBNet converges faster than CNN during





the training period and can reach convergence earlier. Our proposed RRDBNet model has a fast convergence rate and high model efficiency, thus saving computational resources.

**Table 7.** Metrics of extreme precipitation in summer and autumn.

| Models | R95P | | | R99P | | |
|---|---|---|---|---|---|---|
| | Summer (JJA) | | | Summer (JJA) | | |
| | Difference | RMSE | CC | Difference | RMSE | CC |
| GLM | 16.42 | 51.32 | 0.73 | 10.62 | 36.07 | 0.55 |
| CNN | 2.01 | 38.71 | 0.79 | -0.23 | 21.08 | 0.65 |
| RDBNet | 9.52 | 45.51 | 0.72 | 3.19 | 25.27 | 0.58 |
| RRDBNet | -5.44 | 36.65 | 0.80 | -2.15 | 20.42 | 0.65 |
| Models | Autumn (SON) | | | Autumn (SON) | | |
| | Difference | RMSE | CC | Difference | RMSE | CC |
| GLM | 8.09 | 75.61 | 0.23 | 3.67 | 72.99 | 0.05 |
| CNN | 11.56 | 30.71 | 0.54 | 3.50 | 15.52 | 0.21 |
| RDBNet | 9.97 | 29.99 | 0.51 | 3.51 | 16.64 | 0.17 |
| RRDBNet | 4.37 | 22.90 | 0.53 | 1.59 | 13.42 | 0.17 |

**Table 8.** Evaluation metrics of extreme precipitation for each method from 2016 to 2020.

| Models | R95P | | | R99P | | |
|---|---|---|---|---|---|---|
| | Difference | RMSE | CC | Difference | RMSE | CC |
| GLM | 1.37 | 9.57 | 0.86 | 1.10 | 4.42 | 0.82 |
| CNN | 0.15 | 7.75 | 0.89 | 0.03 | 3.28 | 0.85 |
| RDBNet | 0.49 | 9.63 | 0.86 | 0.28 | 4.45 | 0.78 |
| RRDBNet | -1.07 | 7.12 | 0.91 | -0.28 | 3.18 | 0.86 |





**Figure 14.** Seasonal variations of extreme precipitation (mm) in the MRYR from 2016 to 2020 for (a) R95P and (b) R99P.

**Figure 15.** Variations of monthly mean extreme precipitation (mm) in the MRYR during 2016-2020 for (a) R95P and (b) R99P.





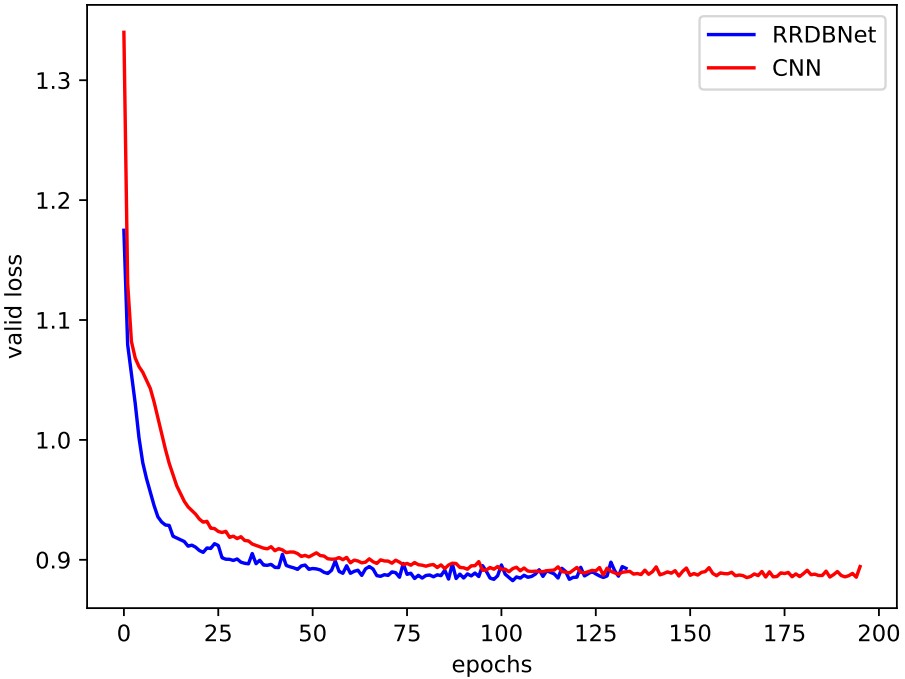

**Figure 16.** Validation loss of RRDBNet and CNN during training.

## 5 Conclusions

In this study, we developed a deep-learning downscaling model (RRDBNet), which was applied to the climate downscaling
over MRYR region. The multilevel residual structure and the idea of dense connectivity help the RRDBNet to adaptively
learn the complex nonlinear relationships between local precipitation and multiple thermodynamic variables. Our results have
comfirmed the applicability of deep learning algorithms in the precipitation downscaling and highlighted the good performance
of our RRDBNet model in extreme precipitation. By systematically evaluating the simulated PDF and extreme precipitation
index including R95P and R99P by observation data, we can infer that RRDBNet is not only greatly improved compared
to GLM in terms of spatial and temporal characteristics of extreme precipitation in the MRYR region, but is also optimal
compared to CNN and RDBNet. Certainly, we have only compared the performance of RRDBNet and CNN in precipitation
downscaling in the MRYR region, and thus, the superior RRDBNet to CNN is actually limited to this region at present. The
applicability of RRDBNet model in other typical climate regions is also required to be evaluated. In addition, the input data for
statistical downscaling in this paper are from ERA5 and not from the GCM simulations. In the future, we will use the simulated
data of GCM to further check these models and predict the future climate change over the MYRY.



*Code and data availability.* All models are based on R software and build on the Climate4R framework (Iturbide et al., 2019). The code for the GLM and CNN models is obtained from https://github.com/SantanderMetGroup/DeepDownscaling (Baño-Medina et al., 2020). To design the RDBNet and RRDBNet used, we rely on downscaleR.keras (https://github.com/SantanderMetGroup/downscaleR.keras). The code for the RDBNet and RRDBNet models is available at https://doi.org/10.5281/zenodo.8234006. The input data from ERA5
(https://doi.org/10.24381/cds.bd0915c6; Hersbach et al., 2023) underlying this study are publicly available. The observational data from GPM (http://gpm.nasa.gov/data-access/downloads/gpm; Huffman et al., 2015) underlying this study are publicly available. The output data can be found at https://doi.org/10.5281/zenodo.8234006.

*Author contributions.* Xiaoning Xie and Cailing Wang designed the study. He Fu did the experiments and wrote the first draft of the paper. Jianing Guo drew the figures in the paper. Chenguang Deng, Heng Liu, Jie Wu, and Zhengguo Shi contributed to the commenting of the
results and revising of the paper.

*Competing interests.* The authors declare that they have no conflict of interest.

*Acknowledgements.* This research has been supported by the National Natural Science Foundation of China (42221003) and the Strategic Priority Research Program of the Chinese Academy of Sciences (XDB40030100). X.N.X. is supported by the National Natural Science Foundation of China (42175059) and the CAS "Light of West China" program (XAB2019A02). Z.G.S. also acknowledges the support of
Youth Innovation Promotion Association CAS (Y2022101) and Natural Science basic Research Program of Shaanxi Province (2022JC-17).





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
