# Peer review of "Deep-learning statistical downscaling of precipitation in the middle reaches of the Yellow River: A Residual in Residual Dense Block based network"

_Geoscientific Model Development, 2023_

## Referee Comment (RC2)

**Deep-learning statistical downscaling of precipitation in the middle reaches of the yellow river: A residual in Residual Dense Block based network:**

The authors have used a residual based approach to downscale precipitation over the reaches of China. The residual-based approach significantly improves on traditional machine learning approaches when compared to observational data.

While there are many aspects of the research that are useful in this manuscript, there is little novelty in this approach and many aspects of the research have been widely investigated elsewhere. The authors have many redundant figures, and the manuscript could be condensed significantly. Additionally, for your evaluation for extremes you only use 4 years of data. I suggest doing another experiment, with a larger test set.

More epxeriments (i.e larger ensemble).

Concerning how much noise there are in the downscaled outputs.

**Title:**

I suggest changing your title to the following or something similar:

*Deep learning-based downscaling of precipitation in the middle reaches of the yellow river using residual networks*

The title is a bit clunky.

**Abstract:**

Line 5: "good performance on precipitation simulations", I'd be explicit and provide some indication of improvement in %. Also, I'd say "when evaluated against observations we obtain X % of improvement relative to linear methods".

**Introduction**

Line 20:" Over simplified parameterizations" I'd remove this text. The main reason for downscaling is the resolution issue. Your research is not on parameterizations.

Line 35 & Line 40: "tricube methods"- what is tricube methods?

Line 45: "Say that capturing non-linearities is important for downscaling variables such as precipitation". This is a good reference (https://www.sciencedirect.com/science/article/pii/S2212094722001049).

Line 50: Bano Medina et al., (2020) is not about forecasting it is about downscaling. If you use that reference, how might need to clarify this.

Line 50: (Pan et al., 2019, Bano Medina 2020) add Sun et al., (2021) and Rampal et al., (2022) . These are also important references. https://www.sciencedirect.com/science/article/pii/S2212094722001049 and https://rmets.onlinelibrary.wiley.com/doi/full/10.1002/joc.6769. You also need to add other recent papers by Bano Medina et al. (e.g. 2021, 2022). I don't think you've summarized the recent literature too well.

Lines 55-60: Here you need to state what contribution your work makes to the overall literature. It's not very convincing about the value added of your research here. Something like "Our work expands on the existing literature by incorporating X y and Z". You could say that we also downscaling the temporal variability which has not been considered before.

**Study Area**

Line 70: Maybe describe why you chose to coarsen ERA5 to $2°$ resolution (i.e. to be consistent with Medina et al., (2020).

**Methodology:**

Lines 85-90: You need to cite other work that as used similar methodology here. For example, Cannon et al., (2008), Rampal et al., (2022), Sun et al., (2021), Bano Medina et al., (2020, 2021, 2022).

Table 1: I personally think this should go in a supplementary section).

Equation 1: Also should go in a supplementary section.

Figure 4: This should go in a supplementary or be combined with Figure 3.

Lines 125: Clarification: is the paper downscaling to daily precipitation? If you are discussing the BG distribution, make sure you cite Rampal et al., (2022) and Sun et al., (2021).

Lines 135: Again are you performing temporal disaggregation? Where for a daily input you are predicting hourly or sub-daily output?

Equation 6: This could be combined in one expression with equation (2).

Lines 170: Are you using any regularization, as these methods can overfit very easily? Please clarify if not.

Lines 180: These metrics are commonly known, so we do not need these in the text. If you'd like to include them add them to the supplementary section or appendix.

**Results:**

Lines 185: This is a very small validation sample (4 years), which makes it challenging to examine the performance on extremes. I'd recommend having a supplementary analysis with a longer validation dataset.

Lines 185: "downscaling projections" – you are not downscaling projections?

Table 2: I think some measure of uncertainty is required perhaps. I'd suggest repeated the experiment 20 times (with different random seeds) and investigate whether your results are statistically significant. This is importance, as this is the premise of your entire results.

Table 4: I don't think table 4 is useful, this could easily go in the supplementary section.

Figure 6: This seems excessive, I suggest making a (2 x 5) plot and combining this with Figure 5. Where the columns are the models (e.g. GPM, GLM) and the row is the climatology and bias.

Figure 5 & 6. It seems a little concerning that there is so much noise in your predictions from the GLM, CNN and other ML models. Other papers do not show such "noise". Are you training your models enough or too much and that your models are overfitting? Some clarification on why there is more noise is needed.

Figure 7: Plot the bias instead of the raw amounts. Reorder axes from DJF, MAM, JJA, SON.

Figure 8: Not useful, this could go in the supplementary instead.

Lines 230: Use either R99P or R95P in the analysis, not both. If you'd like you could keep one in the supplementary section.

Figure 9: Frequency at 100mm is very large in Figure 9b and 9d, is there something wrong in your analysis, this is very concerning?

Figure 10: Again only one of the R95 and R99 plots should be plotted. You should also compute the bias of the R95 fields against the observations (the percentage bias) in one single plot. It seems strange that you have so much noise in your plots.

Figure 11: Again combine with Figure 10.

Table 5: This could be in the supplementary section or combined with information in Figure 10.

Figure 12: Not required, and again very interesting why the outputs are so noisy.

Figure 13: Not required, could be supplementary.

Table 6: Not required, could be supplementary.

Figure 16, should be in the supplementary section.

Figure 15: This is not a commonly used validation metric. I would validate against the RX1Day (wettest day of the year).

Figure 14: This should be the bias instead of the average precipitation.

Table 7 & 8, against too much information, this should be in the supplementary section.

---

## Author Comment (AC2)

Original Manuscript ID: GMD-2023-158

Original Article Title: Deep-learning statistical downscaling of precipitation in the middle reaches of the Yellow River: A Residual in Residual Dense Block based network

Dear Referee #1,

We sincerely thank the Referee #1 for all your valuable comments and insightful suggestions to our manuscript to improve the manuscript. We have addressed all the specific comments in the revised manuscript, with the point-by-point responses detailed below.

Best regards,

Xiaoning Xie et al.

"The authors concluded that RRDBNet performed better than other methods in terms of difference, RMSE and correlartion coefficient (CC). However, the study lacks novelty and improvements are very trivial."

**Response:** Thanks for your comments and suggestions. Our purpose is to explore the performance of deep convolutional neural networks (CNNs) in precipitation downscaling and provide another idea and method for precipitation downscaling. The CNN proposed by Baño-Medina et al. (2020) has only 3 convolutional layers as shown in Figure R1. In the field of computer vision, due to the small number of convolutional layers, on the one hand, these shallow CNNs may not be able to perform in-depth feature extraction and abstraction, and may not adequately capture complex patterns and relationships in the data; on the other hand, shallow CNNs do not have the ability to extract features at more hierarchical levels, and may be difficult to fully learn higher-level, more abstract, and more discriminative feature information and are easier to produce overfitting (Simonyan and Zisserman, 2015; Szegedy et al., 2015; LeCun et al., 2015; He et al., 2016; Huang et al., 2017). Therefore, in this paper, we explore the performance of deep CNNs in precipitation downscaling to provide another idea and method for precipitation downscaling. Here, we develop a deep learning-based statistical downscaling method, i.e., Residual-in-Residual Dense Block based network (RRDBNet) model (Figure R2), to produce daily high-resolution precipitation over the MRYR region. Residualin-Residual Dense Block (RRDB) is introduced in the proposed RRDBNet to optimize the structure of the network model. The multi-level residuals and dense connectivity strategies in RRDB help the model to adequately extract more abstract and advanced feature information (He et al., 2016; Huang et al., 2017; Zhang et al., 2018; Wang et al., 2018). Extensive experiments in this paper show that the proposed RRDBNet exhibits excellent performance in most metrics. This indicates that the proposed RRDBNet is effective and can be used as a reliable tool for precipitation downscaling.

Specifically, the proposed RRDBNet reduces RMSE by 58% (79%) and improves CC by 38% (145%) relative to GLM on climatology mean for R95P (R99P). It is also the best compared to other deep-learning models, with the smallest RMSE and the largest CC. In terms of temporal variation in annual extreme precipitation, RRDBNet decreases RMSE by 28% (49%) and enhances CC by 229% (355%) relative to GLM for R95P (R99P), and also has significant improvements over CNN and RDBNet (see Supplementary Table 4). Our results indicate the applicability of deep-learning algorithms in the precipitation downscaling and highlight the good performance of RRDBNet in extreme precipitation. The corresponding descriptions have been added in the revised version.

[Figure]

Figure R1. The CNN overall framework (adapted by Baño-Medina et al. (2020)).

(a) Framework of RRDBNet

[Figure]

(b) Dense Block

Figure R2. RRDBNet overall framework.

**Comment 1:** The models were trained with daily data, but all the evaluations were performed at aggregated time scale (annual, monthly and seasons). How does the model perform in daily time scale (both daily statistics and extremes)? The aggregated time scales may not be critical and may hide important information on model evaluation.

**Author response:** Thanks for your comments and suggestions. In the precipitation downscaling of this paper, we use probabilistic (stochastic) modeling (Cannon, 2008; Baño-Medina et al., 2020, 2022; Sun and Lan, 2021; Rampal et al., 2022) rather than deterministic modeling (Bürger, 1996, 2002). Deterministic modeling often leads to an underestimation of the variability and the extremes because the explained variance may be significantly smaller than the observed one (Williams, 1997; Cannon, 2008; Baño-Medina et al, 2020, 2022). This is particularly relevant for precipitation, whose local variability is often influenced by local phenomena that are not captured by the selected predictors (Schoof and Pryor, 2001; Maraun and Widmann, 2018). We also compare the effect of deterministic and stochastic modeling of precipitation on the probability density function (pdf) of daily precipitation frequency in Figure R3. It is obvious that deterministic precipitation modeling does not conform to the changing trend of observed precipitation frequency, while stochastic precipitation modeling is roughly

consistent with the changing trend of observed precipitation frequency. Probabilistic modeling downscaling yields daily precipitation as random values derived from the Bernoulli-Gamma distribution. It may be more suitable for climatological statistical analysis. In the future we use the model also to do the assessment of future centennial precipitation changes in the region. In addition, based on your suggestions, we have done the validation of the daily precipitation in the area as shown in Table R1. We found that the RRDBNet model also performs well on daily precipitation compared to other models, having the smallest RMSE and largest correlation coefficient (CC). Since the five-year daily data we obtained after downscaling is $1827 \times 100 \times 159$, when we use the five-year daily data to draw a scatter plot, the output is very slow. Therefore, we did a regional average before drawing the corresponding scatterplot (Figure R4). It can be seen that also RRDBNet performs well.

[Figure]

Figure R3. (a) Comparison of probability density functions of daily precipitation from 1 mm to 100 mm for all grids of GPM, and RRDBNet (stochastic and deterministic) in the MRYR from 2016 to 2020. (b) The same as (a) but GPM, and CNN (stochastic and deterministic).

Table R1. Evaluation metrics of daily precipitation in different models

| Models | Difference | RMSE | CC |
|--------|-----------|------|-----|
| GLM | -0.12 | 11.65 | 0.17 |
| CNN | 0.07 | 6.30 | 0.39 |
| RDBNet | 0.03 | 6.55 | 0.37 |
| RRDBNet | -0.10 | 5.93 | 0.40 |

[Figure]

Figure R4. Scatter plot of daily precipitation after regional averaging in the MRYR from 2016 to 2020 for (a) GLM, (b) CNN, (c) RDBNet, and (d) RRDBNet.

**Comment 2:** The rationale of the proposed method is not clear. Compared to other deep learning methods, what are its advantages and why did the author propose this method? Without deep understanding the model itself, the manuscript gives readers limited insights.

**Author response:** Thanks for your comments and suggestions. To describe the rationale of the proposed RRDBNet more clearly, we have made corresponding changes to the methodological introduction in the revised manuscript (see Section 3). We have also explained the purpose and

advantages of the proposed method in Instruction of the revised manuscript. Our purpose is to explore the performance of deep convolutional neural networks (CNNs) in precipitation downscaling and provide another idea and method for precipitation downscaling. The CNN proposed by Baño-Medina et al. (2020) has only 3 convolutional layers. In the field of computer vision, due to the small number of convolutional layers, on the one hand, these shallow CNNs may not be able to perform in-depth feature extraction and abstraction, and may not adequately capture complex patterns and relationships in the data; on the other hand, shallow CNNs do not have the ability to extract features at more hierarchical levels, and may be difficult to fully learn higher-level, more abstract, and more discriminative feature information and are easier to produce overfitting (Simonyan and Zisserman, 2015; Szegedy et al., 2015; LeCun et al., 2015; He et al., 2016; Huang et al., 2017). Therefore, in this paper, we explore the performance of deep CNNs in precipitation downscaling to provide another idea and method for precipitation downscaling. we develop a deep learning-based statistical downscaling method, i.e., Residual-in-Residual Dense Block based network (RRDBNet) model, to produce daily high-resolution precipitation over the MRYR region. Residual-in-Residual Dense Block (RRDB) is introduced in the proposed RRDBNet to optimize the structure of the network model. The multi-level residuals and dense connectivity strategies in RRDB help the model to adequately extract more abstract and advanced feature information (He et al., 2016; Huang et al., 2017; Zhang et al., 2018; Wang et al., 2018). Extensive experiments in this paper show that the proposed RRDBNet exhibits excellent performance in most metrics. This indicates that the proposed RRDBNet is effective and can be used as a reliable tool for precipitation downscaling.

**Comment 3:** The authors claimed the proposed method is much better than the other three methods, which may not be true. Given the stochastic nature of deep learning models and the slightly better statistics, the superiority may be purely due to stochasticity itself. Training the model multiple times may help discriminate the superiority and stochasticity. Furthermore, it is not fair to compare different deep learning models without giving model complexity (e.g., the number of trainable parameters).

**Author response:** We sincerely appreciate your valuable comments and suggestions. We aim to study the performance of deep CNNs in precipitation downscaling and provide another idea

and method for precipitation downscaling. In this paper, we have conducted a large number of experiments. These experiments show that the proposed RRDBNet outperforms other models on most metrics, and not only on a single metric. We also repeated the experiment ten times for RRDBNet, and calculated the mean and variance of the ten experimental results in Tables R2 and R3. We found that the changes in the results of the ten experiments are very small and all outperformed other models. These may indicate that RRDBNet is an effective tool for downscaling precipitation. Furthermore, from the model structure analysis, the CNN proposed by Baño-Medina et al. (2020) has only three convolutional layers (Figure R1) and is a shallow CNN. Whereas, the multilevel residual and densely connected structure of RRDBNet (Figure R2) has been proven to capture more advanced, more abstract, and more discriminative features and more complex nonlinear relationships than these shallow CNNs in the field of computer vision (Simonyan and Zisserman, 2014; Szegedy et al., 2015; LeCun et al., 2015; He et al., 2016; Huang et al., 2017). We also compare the convergence of RRDBNet and CNN during training in subsection 4.4 of the original manuscript. The result is shown in Figure R5. We found that under the same experimental equipment and experimental parameter configuration, RRDBNet converges faster than CNN. Faster model convergence can save computational resources and time during training.

Table R2. Results of ten repeated experiments using RRDBNet.

|  | Difference | RMSE | CC |
|---|---|---|---|
| 1 | -0.09589823 | 0.29287103 | 0.93531325 |
| 2 | -0.0973649 | 0.29267051 | 0.93549379 |
| 3 | -0.09529275 | 0.29251509 | 0.93530962 |
| 4 | -0.09570838 | 0.29507643 | 0.93433473 |
| 5 | -0.09507929 | 0.29269518 | 0.93511726 |
| 6 | -0.09596082 | 0.2931742 | 0.93505923 |
| 7 | -0.09697484 | 0.29216068 | 0.93575122 |
| 8 | -0.09673768 | 0.29315587 | 0.93529157 |
| 9 | -0.09631896 | 0.29314359 | 0.93520193 |
| 10 | -0.09771557 | 0.29449235 | 0.93481579 |
| Ensemble | -0.0963±0.0008 | 0.2932±0.0009 | 0.9352±0.0004 |

Table R3. Evaluation metrics of annual mean precipitation in different models.

| Models | Difference | RMSE | CC |
|---|---|---|---|
| GLM | -0.12 | 0.36 | 0.89 |
| CNN | 0.07 | 0.35 | 0.92 |
| RDBNet | 0.03 | 0.33 | 0.92 |
| RRDBNet | -0.10 | 0.29 | 0.94 |
| RRDBNet(10 times) | -0.0963±0.0008 | 0.2932±0.0009 | 0.9352±0.0004 |

[Figure]

Figure R5. Validation loss of RRDBNet and CNN during training.

**Comment 4:** In the introduction section, the authors described many GCM downscaling. However, this study is not about GCM downscaling but reanalysis data, which is very different story and may mislead readers. Thus, the introduction needs to be rewritten.

**Author response:** Thanks for your nice suggestions. Different statistical downscaling (Maraun and Widmann, 2018) methods have been developed building on empirical relationships

established between informative large-scale atmospheric variables (predictors) and local/regional variables of interest (predictands). Under the perfect-prognosis approach, these relationships are learned from (daily) data using simultaneous observations for both the predictors (from a reanalysis) and predictands (historical local or gridded observations), and are subsequently applied to GCM-simulated predictors (multidecadal climate change projections under different scenarios), to obtain locally downscaled values (see, e.g., Gutiérrez et al, 2013; Manzanas et al, 2018; Baño-Medina et al., 2020, 2022). Therefore, we first use RRDBNet to learn this statistical relationship from reanalysis data ($2^{\circ} \times 2^{\circ}$) and observation data ($0.1^{\circ} \times 0.1^{\circ}$), and then use the trained RRDBNet to act on the predictors ($2^{\circ} \times 2^{\circ}$) of GCM in the future. We have made instructions in Section 2 of the revised manuscript. Currently, we have also done experiments to use the trained RRDBNet for the GCM downscaling. The results of the experiment are shown in Figure R6. In variations of annual mean precipitation, the precipitation (MPI-ESM1-2-LR ($0.1^{\circ} \times 0.1^{\circ}$)) after the downscaling of RRDBNet is more consistent with the observation (CN05) than the precipitation of GCM (MPI-ESM1-2-LR ($2^{\circ} \times 2^{\circ}$)). This also proves that our RRDBNet is an effective and reasonable tool for downscaling precipitation.

The corresponding descriptions in the revised manuscript are as follows: **Under the "perfect-prognosis", the model learns statistical relationships from daily predictors (from reanalysis data) and predictands (from historical observations), and then works on the predictors of the GCM to obtain the corresponding regional or local downscaling results (Gutiérrez et al., 2013; Manzanas et al., 2018; Baño-Medina et al., 2020, 2022). Therefore, we selected predictors from ERA5 and used cumulative precipitation from Integrated Multi-satellite Retrievals for Global Precipitation Measurement (IMERG) as predictand.**

[Figure]

Figure R6. Spatial distribution of annual mean precipitation (mm/day) in the MRYR from 1501 to 2000 for (a) CN05, (b) MPI-ESM1-2-LR (2°×2°), and (c) MPI-ESM1-2-LR (0.1°×0.1°). (d) Variations of annual mean precipitation (mm/day) in the MRYR during 1501-2000 for CN05, MPI-ESM1-2-LR (2°×2°), and MPI-ESM1-2-LR (0.1°×0.1°).

**Comment 5:** Line 69: the reanalysis data ERA5 has spatial resolution of 0.25x0.25 degree not 2x2 degree. Furthermore, the authors selected 5 predictors without giving any reasons.

**Author response:** We sincerely appreciate your valuable comments. Note that most GCMs in CMIP5/6 have coarser resolution (almost 2 degree), we are going to downscale the GCM precipitation in the future. Therefore, we choose the ERA5 predictors ($2° × 2°$) to train the model, which is upscaled from the default resolution 0.25×0.25 degree (as suggested by Baño-Medina et al. (2020)). The ERA5 data with $2° × 2°$ can also be directly downloaded from https://www.ecmwf.int/en/forecasts/dataset/ecmwf-reanalysis-v5. These selected five predictors (Q, T, Z, U, and V) generally contain all the most of information related to the surface precipitation, which is consistent with the previous studies (Baño-Medina et al., 2020, 2022; Sun and Lan, 2022; Rampal et al., 2022). In addition, to facilitate the comparison between the proposed RRDBNet and the CNN proposed by Baño-Medina et al. (2020), so we choose the predictors selected by. i.e. the 5 predictors described in this paper. We have also clarified this point in Section 2 of the revised manuscript. The clarifications are as follows: **Specifically, the**

input dataset (predictor set) of statistic downscaling is derived from data with $2^{\circ} \times 2^{\circ}$ resolution in ERA5, which is upscaled from the default resolution $0.25 \times 0.25$ degree (consistent with the predictors selected by Baño-Medina et al. (2020)).

**Comment 6:** Line 89: the three parameters came out first time without any explanations. Y $\in$ Rt°×100°×159 came out without further information.

**Author response:** Thanks for your great suggestions on improving our manuscript. Following your suggestion, we have made the explanation when these three parameters (p, $\alpha$, and $\beta$) were first mentioned in Subsection 3.1 of the revised manuscript. $Y \in \mathbb{R}^{t \times 100 \times 159}$ should be written as $Y \in \mathbb{R}^{100 \times 159}$. We have fixed this error in Subsection 3.1 of the revised manuscript. $Y$ represents the studied precipitation area. $100 \times 159$ is the grid matrix after precipitation downscaling. The modification is as follows: **Overall, the predictors are processed by the statistical downscaling model (RRDBNet) to obtain the modeled parameters p, α, and β. Then precipitation is estimated using a mixed binomial-lognormal distribution with modeled p, α, and β. This is consistent with Baño-Medina et al., (2020, 2022), Sun et al., (2021), and Rampal et al., (2022). p denotes the probability of precipitation, and α and β represent the shape and scale of the gamma distribution, respectively. Finally, the precipitation $Y \in \mathbb{R}^{100 \times 159}$ is obtained by calculating the parameters p, α, and β (Baño-Medina et al., 2020, 2022; Sun and Lan, 2021; Rampal et al., 2022).**

**Comment 7:** Lines 114 to 119: Does the statement about batch normalization come from model testing? If that is true, this statement needs to be included in the results section. If it is not true, where this come from?

**Author response:** Thank you very much for your comments. We originally meant that there is no Batch Normalization (BN) layer (Ioffe and Szegedy, 2015) in the employed Residual-in-Residual Dense Block (RRDB) structure (Wang et al., 2018). The reason is that the BN layer may introduce artifacts that limit the generalization ability of the model when the statistics of the model's training and test sets are different. Therefore, there is no BN layer in our proposed RRDBNet. To avoid ambiguity and better express our meaning, we have deleted this content in the third paragraph of subsection 3.1.1 of the revised manuscript. In this paper, both training

and test data are standardized before being input into the model.

**Comment 8:** Line 121: how did you get B=0.2?

**Author response:** Thank you very much for your comment. We set B to 0.2 based on the paper by Wang et al. (2018). We have added the corresponding reference for clarification in Subsection 3.1.1 of the revised manuscript. The modification is as follows: **in this paper we set B=0.2 (Wang et al., 2018).**

**Comment 9:** Line 139: The statement "The final precipitation is obtained by multiplying p with the random values of the distribution with shape α scale β." Why?

**Author response:** Thank you for your comments. Due to the mixed discrete-continuous nature of precipitation, the model predicts the precipitation by modelling the parameters of Bernoulli-Gamma distribution (Williams, 1997; Cannon, 2008; Baño-Medina et al., (2020, 2022), Sun et al., (2021), and Rampal et al., (2022)). The precipitation estimated through a mixed binomial-lognormal distribution of which the corresponding parameters (p, α, β) are modelled (Baño-Medina et al., (2020, 2022), Sun et al., (2021), and Rampal et al., (2022)). We have made an explanation of this in Subsection 3.1 of the revised manuscript. The explanation is as follows: **Overall, the predictors are processed by the statistical downscaling model (RRDBNet) to obtain the modeled parameters p, α, and β. Then precipitation is estimated using a mixed binomial-lognormal distribution with modeled p, α, and β. This is consistent with Baño-Medina et al., (2020, 2022), Sun et al., (2021), and Rampal et al., (2022). p denotes the probability of precipitation, and α and β represent the shape and scale of the gamma distribution, respectively. Finally, the precipitation $Y \in \mathbb{R}^{100 \times 159}$ is obtained by calculating the parameters p, α, and β (Baño-Medina et al., 2020, 2022; Sun and Lan, 2021; Rampal et al., 2022).**

**Comment 10:** Line 142: "In the training phase" was repeated.

**Author response:** Thanks for pointing out the mistake. We have modified it in Subsection 3.1.3 of the revised manuscript. The modification is as follows: **In the training phase, when the model outputs predicted values by forward propagation, the loss function calculates the**

**difference between the predicted and actual values, i.e., the loss value.**

References

Baño-Medina, J., Manzanas, R., and Gutiérrez, J. M.: Configuration and intercomparison of deep learning neural models for statistical downscaling, Geosci. Model Dev., 13, 2109–2124, 2020.

Baño-Medina, J., Manzanas, R., Cimadevilla, E., Fernández, J., González-Abad, J., Cofiño, A. S., and Gutiérrez, J. M.: Downscaling multi-model climate projection ensembles with deep learning (DeepESD): contribution to CORDEX EUR-44, Geosci. Model Dev., 15, 6747–6758, 2022.

Bürger, G.: Expanded downscaling for generating local weather scenarios, Clim. Res., 7, 111–128, 1996.

Bürger, G.: Selected precipitation scenarios across Europe, J. Hydrol., 262, 99–110, 2002.

Cannon, A. J.: Probabilistic Multisite Precipitation Downscaling by an Expanded Bernoulli-Gamma Density Network, J. Hydrometeorol., 9, 1284–1300, 2008.

Gutiérrez, J. M., San-Martín, D., Brands, S., Manzanas, R., and Herrera, S.: Reassessing Statistical Downscaling Techniques for Their Robust Application under Climate Change Conditions, J. Clim., 26, 171–188, 2013.

He, K., Zhang, X., Ren, S., and Sun, J.: Deep residual learning for image recognition, in: Proc IEEE Comput Soc Conf Comput Vision Pattern Recognit, pp. 770–778, 2016.

Huang, G., Liu, Z., Van Der Maaten, L., and Weinberger, K. Q.: Densely connected convolutional networks, in: Proc IEEE Comput Soc Conf Comput Vision Pattern Recognit, pp. 4700–4708, 2017.

Ioffe, S. and Szegedy, C.: Batch normalization: Accelerating deep network training by reducing internal covariate shift, in: Int. Conf. Mach. Learn., ICML, pp. 448–456, pmlr, 2015.

LeCun, Y., Bengio, Y., and Hinton, G.: Deep learning, nature, 521, 436–444, 2015.

Manzanas, R., Lucero, A., Weisheimer, A., & Gutiérrez, J. M. (2018). Can bias correction and statistical downscaling methods improve the skill of seasonal precipitation forecasts?. Climate dynamics, 50, 1161-1176.

Maraun, D. and Widmann, M.: Statistical Downscaling and Bias Correction for Climate

Research, Cambridge University Press, 2018.

Rampal, N., Gibson, P. B., Sood, A., Stuart, S., Fauchereau, N. C., Brandolino, C., Noll, B., and Meyers, T.: High-resolution downscaling with interpretable deep learning: Rainfall extremes over New Zealand, Weather and Climate Extremes, 38, 100 525, 2022.

Schoof, J. T. and Pryor, S.: Downscaling temperature and precipitation: a comparison of regression-based methods and artificial neural networks, Int. J. Climatol., 21, 773–790, 2001.

Simonyan, K. and Zisserman, A.: Very deep convolutional networks for large-scale image recognition, 2015.

Sun, L. and Lan, Y.: Statistical downscaling of daily temperature and precipitation over China using deep learning neural models: Localization and comparison with other methods, International Journal of Climatology, 41, 1128–1147, 2021.

Szegedy, C., Liu, W., Jia, Y., Sermanet, P., Reed, S., Anguelov, D., Erhan, D., Vanhoucke, V., and Rabinovich, A.: Going deeper with convolutions, vol. 07-12-June-2015, pp. 1 – 9, 2015.

Wang, X., Yu, K., Wu, S., Gu, J., Liu, Y., Dong, C., Qiao, Y., and Change Loy, C.: ESRGAN: Enhanced super-resolution generative adversarial networks, in: Lect. Notes Comput. Sci., pp. 0–0, 2018.

Williams, P.: Modelling Seasonality and Trends in Daily Rainfall Data, NIPS, 10, 1997.

Zhang, Y., Tian, Y., Kong, Y., Zhong, B., and Fu, Y.: Residual Dense Network for Image Super-Resolution, in: Proc. IEEE Comput. Soc. Conf. Comput. Vision Pattern Recognit, pp. 2472–2481, 2018.

---

## Author Comment (AC3)

Original Manuscript ID: GMD-2023-158

Original Article Title: Deep-learning statistical downscaling of precipitation in the middle reaches of the Yellow River: A Residual in Residual Dense Block based network

Dear Referee #2,

We sincerely thank you for all your valuable comments, insightful suggestions, and thoughtful corrections to our manuscript. These comments and suggestions will undoubtedly help us improve the quality of the manuscript. Below are our point-by-point responses to the comments. In the revised manuscript, all changes and additions are highlighted in yellow.

Best regards,

Xiaoning Xie et al.

"The authors have many redundant figures, and the manuscript could be condensed significantly. Additionally, for your evaluation for extremes you only use 4 years of data. I suggest doing another experiment, with a larger test set. More epxeriments (i.e larger ensemble). Concerning how much noise there are in the downscaled outputs."

**Response:** Thanks for your comments and suggestions. Based on your suggestions, we have significantly condensed the manuscript, merged some figures and tables, and put some figures and tables in the Supplementary Section in the revised manuscript. We also did the experiments to extend the testing period. Specifically, For RRDBNet, we did the test experiments for 10 years (2011-2020), and the training period was 2001-2011. The spatial distribution is shown in Figure R1. For the ten-year (2011-2020) test period, we calculated the specific metrics for the deep-learning models in Table R1. As noted by the Referee, when we increase the time of validation for 10 years (2011-2020), the noise decreases significantly as in Figure R1. As can be seen from Table 1, for the ten-year test period (2011-2020), RRDBNet outperforms other models on R95P, R99P, and RX1Day (with the smallest RMSE and largest CC). Therefore, RRDBNet has good performance in capturing extreme precipitation.

In addition, for RRDBNet, we repeated the experiment as ten times and calculated the mean and variance for these experiment results in Tables R2 and R3. We found that the changes in the results of these experiments are very small in Table R2. Then, we calculated the mean and variance of these results in Table R2 and Table R3. Therefore, we infer that the RRDBNet results are statistically significant in Table R3, compared with other models. We have analyzed the above results in the Discussion Section of the revised manuscript.

[Figure]

Figure R1. Spatial distribution of annual mean precipitation and extreme precipitation in the MRYR for RRDBNet in different testing periods.

Table R1. Evaluation metrics of annual extreme precipitation for each method from 2011 to 2020

|  | Models | Difference | RMSE | CC |
|---|---|---|---|---|
|  | CNN | 4.84 | 28.80 | 0.56 |
| R95P | RDBNet | 1.84 | 26.99 | 0.62 |
|  | RRDBNet | -3.62 | 25.37 | 0.66 |
|  | CNN | 3.97 | 15.41 | 0.37 |
| R99P | RDBNet | 2.57 | 13.63 | 0.48 |
|  | RRDBNet | 0.89 | 12.81 | 0.51 |
|  | CNN | 8.65 | 11.09 | 0.32 |
| RX1Day | RDBNet | 6.57 | 8.44 | 0.41 |
|  | RRDBNet | 4.86 | 6.76 | 0.55 |

Table R2. Results of ten repeated experiments using RRDBNet.

|  | Difference | RMSE | CC |
|---|---|---|---|
| 1 | -0.09589823 | 0.29287103 | 0.93531325 |
| 2 | -0.0973649 | 0.29267051 | 0.93549379 |
| 3 | -0.09529275 | 0.29251509 | 0.93530962 |
| 4 | -0.09570838 | 0.29507643 | 0.93433473 |
| 5 | -0.09507929 | 0.29269518 | 0.93511726 |
| 6 | -0.09596082 | 0.2931742 | 0.93505923 |
| 7 | -0.09697484 | 0.29216068 | 0.93575122 |
| 8 | -0.09673768 | 0.29315587 | 0.93529157 |
| 9 | -0.09631896 | 0.29314359 | 0.93520193 |
| 10 | -0.09771557 | 0.29449235 | 0.93481579 |
| Ensemble | -0.0963±0.0008 | 0.2932±0.0009 | 0.9352±0.0004 |

Table R3. Evaluation metrics of annual mean precipitation in different models.

| Models | Difference | RMSE | CC |
|---|---|---|---|
| GLM | -0.12 | 0.36 | 0.89 |
| CNN | 0.07 | 0.35 | 0.92 |
| RDBNet | 0.03 | 0.33 | 0.92 |
| RRDBNet | -0.10 | 0.29 | 0.94 |
| RRDBNet(10 times) | -0.0963±0.0008 | 0.2932±0.0009 | 0.9352±0.0004 |

**Comment 1:** I suggest changing your title to the following or something similar: Deep learning-based downscaling of precipitation in the middle reaches of the yellow river using residual networks. The title is a bit clunky.

**Author response:** Thanks for your great suggestions on improving our manuscript. Based on your suggestions, we have modified the title to "Deep learning-based downscaling of precipitation in the middle reaches of the Yellow River using Residual-in-Residual Dense Block based networks".

**Comment 2:** Line 5: "good performance on precipitation simulations", I'd be explicit and provide some indication of improvement in %. Also, I'd say "when evaluated against observations we obtain X % of improvement relative to linear methods".

**Author response:** Thanks for your great suggestion on improving our manuscript. We have included specific indicators of improvement in the abstract of the revised manuscript. The modification is as follows: **The results show that the proposed RRDBNet has a good performance on precipitation simulations, which can well reproduce the spatial-temporal characteristics of high-resolution precipitation. RRDBNet reduces RMSE by 19% and improves CC by 6% relative to GLM for climatology mean. Especially, RRDBNet has substantial improvements in extreme precipitation compared with other models. It reduces RMSE by 58% (79%) and improves CC by 38% (145%) relative to GLM for R95P (R99P).**

**Comment 3:** Line 20:" Over simplified parameterizations" I'd remove this text. The main reason for downscaling is the resolution issue. Your research is not on parameterizations.

**Author response:** We sincerely appreciate your valuable comments. Based on your suggestion we have removed "Over simplified parameterizations" in the introduction. We have modified this sentence the revised manuscript as follows: **Due to the coarse spatial resolutions (usually ~100 km) in Global Climate Models (GCMs) (Berg et al., 2013)**

**Comment 4:** Line 35 & Line 40: "tricube methods"- what is tricube methods?

**Author response:** Thanks for your comments. The tricube method is a statistical method of non-parametric regression used to estimate the smoothing curve of the data (Wand et al., 1994; Cleveland et al., 2017). It estimates the value of an unknown data point by weighted smoothing a set of neighboring data points around each data point. The method uses a cubic spline function for smoothing, so it is called a tricube (three times cubed) method. In the tricube method, the data points around the point to be interpolated are organized into a cube, i.e., a three-dimensional space. Then, different weights are given to each data point within the cube according to its distance from the point to be interpolated. Typically, data points that are closer to the point to be interpolated are given higher weights, while data points that are farther away are given lower weights. The advantage of this method is that the spatial distribution of data points can be considered during the interpolation process, and data points that are far away from the point to be interpolated are given lower weights to reduce their impact on the interpolation results. This makes the tricube method widely used in the field of precipitation. Such as estimation of future precipitation quantiles (Stojkovic et al., 2019), bias correction of GCM data (Panjwani et al., 2021) and estimation of missing daily rainfall data (Makungo et al., 2019).

**Comment 5:** Line 45: "Say that capturing non-linearities is important for downscaling variables such as precipitation". This is a good reference (https://www.sciencedirect.com/science/article/pii/S2212094722001049).

**Author response:** Thanks for your great suggestion on improving our manuscript. We have added this reference to the introduction based on your suggestion and have modified this sentence appropriately. The modification is as follows: **Although the traditional methods are**

easy to understand and interpret, they cannot accurately characterize the nonlinear dependence among climate variables due to their simple assumptions (Booij, 2002; Beniston et al., 2007; Prudhomme and Davies, 2009), and capturing non-linearities is very important for downscaling variables such as precipitation (Rampal et al., 2022).

**Comment 6:** Line 50: Bano Medina et al, (2020) is not about forecasting it is about downscaling. If you use that reference, how might need to clarify this.

**Author response:** Thank you for pointing out the mistakes. We have made corrections in the revised manuscript. The modification is as follows: **To improve the quality of downscaling, many models about CNNs are employed to generate high-resolution data (Rodrigues et al., 2018; Pan et al., 2019; Baño-Medina et al., 2020; Sun and Lan, 2021; Rampal et al., 2022).**

**Comment 7:** Line 50: (Pan et al, 2019, Bano Medina 2020) add Sun et al, (2021) and Rampal et al, (2022). These are also important references. https://www.sciencedirect.com/science/article/pii/S2212094722001049 and https://rmets.onlinelibrary.wiley.com/doi/full/10.1002/joc.6769. You also need to add other recent papers by Bano Medina et al (e.g. 2021, 2022). I don't think you've summarized the recent literature too well.

**Author response:** We sincerely appreciate your valuable comments. Based on your suggestion, we have added references (Sun and Lan, 2021; Rampal et al., 2022) and rewritten this part in the revised manuscript. The rewrite is as follows: **It was demonstrated that CNNs have better performance on the estimation of precipitation than reanalysis products and statistical downscaling products obtained using linear regression, nearest neighbor and random forest (Pan et al., 2019; Baño-Medina et al., 2020; Sun and Lan, 2021; Rampal et al., 2022). For example, experimental results from Pan et al. (2019) at 14 geogrid points in the U.S. show that CNN-based precipitation estimation outperforms reanalyzed precipitation and downscaled precipitation estimation using linear regression, nearest neighbor, and random forest. Baño-Medina et al. (2020) conducted precipitation downscaling experiments in continental Europe, and their experiments showed that the overall**

performance of the CNN is superior to the generalized linear regression model (GLM) for precipitation. Later, Baño-Medina et al. (2022) used CNN for GCM downscaling. They emphasized that CNN-based downscaling results can reproduce the spatial distribution of precipitation and temperature observations during historical periods and reduce the systematic biases shown by global and regional physical models. In future climate change analysis, the spatial pattern and magnitude of CNN-based downscaling are roughly similar to RCM. Sun et al. (2021) further demonstrated that the performance of CNN for precipitation downscaling is superior to GLM in China, and CNN outperforms in large regional downscaling. Rampal et al. (2022) investigated learning relationships in CNN by implementing gradient-weighted class-activation maps (Grad-CAM), and they demonstrated that CNN can automatically learn physically believable relationships between large-scale atmospheric environments and extreme localized precipitation events.

**Comment 8:** Lines 55-60: Here you need to state what contribution your work makes to the overall literature. It's not very convincing about the value added of your research here. Something like "Our work expands on the existing literature by incorporating X y and Z". You could say that we also downscaling the temporal variability which has not been considered before.

**Author response:** Thanks for your comments and suggestions. According to your suggestion, We have stated the contribution of our work to the overall literature in the revised manuscript. The statement is as follows: The main contributions of our work are as follows: **1. We developed an RRDB-based network to provide another idea and method for precipitation downscaling. Multi-level residuals and dense connectivity strategies are incorporated into the RRDB, which is beneficial to extracting higher-level, more abstract, and more discriminative features in atmospheric variables, capturing complex patterns and nonlinear features among climate variables, and enhancing model stability. 2. The proposed RRDBNet more accurately captures the extreme precipitation and extreme precipitation frequency, and has better results in terms of extreme precipitation compared to other models. 3. We also downscaled temporal variability not previously considered by** Baño-Medina et al (2020).

**Comment 9:** Line 70: Maybe describe why you chose to coarsen ERA5 to 2° resolution (i.e. to be consistent with Medina et al, (2020).

**Author response:** Thank you for your suggestions. Under the perfect-prognosis approach, the statistical downscaling relationships are learned from (daily) data using simultaneous observations for both the predictors (from a reanalysis) and predictands (historical local or gridded observations), and are subsequently applied to GCM-simulated predictors (multidecadal climate change projections under different scenarios), to obtain locally downscaled values (Gutiérrez et al., 2013; Manzanas et al., 2018; Baño-Medina et al., 2020, 2022). Therefore, we choose data from ERA5 as the predictors and cumulative precipitation from Integrated Multi-satellite Retrievals for Global Precipitation Measurement (IMERG) as the predictand. To facilitate comparison of performance between models, the spatial resolution of the predictors was $2° \times 2°$ (consistent with the spatial resolution of the predictors chosen by Baño-Medina et al. (2020)). In addition, in the future, we will use the trained model to downscale the $2° \times 2°$ GCM predictors. We have described these in Section 2 of the revised manuscript. The description is as follows: **Under the "perfect-prognosis", the model learns statistical relationships from daily predictors (from reanalysis data) and predictands (from historical observations), and then works on the predictors of the GCM to obtain the corresponding regional or local downscaling results (Gutiérrez et al., 2013; Manzanas et al., 2018; Baño-Medina et al., 2020, 2022). Therefore, we selected predictors from ERA5 and used cumulative precipitation from Integrated Multi-satellite Retrievals for Global Precipitation Measurement (IMERG) as predictand. Specifically, the input dataset (predictor set) of statistic downscaling is derived from data with $2° \times 2°$ resolution in ERA5 (consistent with the predictors selected by Baño-Medina et al. (2020)).**

**Comment 10:** Lines 85-90: You need to cite other work that as used similar methodology here. For example, Cannon et al, (2008), Rampal et al, (2022), Sun et al, (2021), Bano Medina et al, (2020, 2021, 2022).

**Author response:** Thanks for your suggestions and comments. We have cited related work using similar methodologies in Subsection 3.2 of the revised manuscript. The change is as

follows: **Overall, the predictors are processed by the statistical downscaling model (RRDBNet) to obtain the modeled parameters p, α and β. Then precipitation is estimated using a mixed binomial-log-normal distribution with modeled p, α and β. This is consistent with Baño-Medina et al., (2020, 2022), Sun et al., (2021), and Rampal et al., (2022). p denotes the probability of precipitation, and α and β represent the shape and scale of the gamma distribution, respectively.**

**Comment 11:** Table 1: I personally think this should go in a supplementary section.

**Author response:** Thanks for your great suggestion on improving our manuscript. We have placed this table in the Supplementary Section and noted it as Supplementary Table 1.

**Comment 12:** Equation 1: Also should go in a supplementary section.

**Author response:** Thanks for your suggestion. Following your suggestion, we have also put this equation in the Supplementary Section and noted it as Supplementary Equation 1.

**Comment 13:** Figure 4: This should go in a supplementary or be combined with Figure 3.

**Author response:** Thanks for your comment. We have put this figure in the Supplementary Section and recorded it as Supplementary Figure 1.

**Comment 14:** Lines 125: Clarification: is the paper downscaling to daily precipitation? If you are discussing the BG distribution, make sure you cite Rampal et al, (2022) and Sun et al, (2021).

**Author response:** Thank you very much for your comments. According to your suggestions, we have clarified that we study downscaled daily precipitation and have added these two references (Sun and Lan, 2021; Rampal et al., 2022) in the revised manuscript. The modification is as follows: **Therefore, we study stochastic prediction and downscaled daily precipitation in this paper. Due to the mixed discrete and continuous nature of precipitation, Williams (1997) suggested using the bernoulli-gamma distribution to describe precipitation, which has been used for single-site precipitation downscaling models (Haylock et al., 2006; Cawley et al., 2007) and gridded precipitation downscaling models (Baño-Medina et al., 2020, 2022; Sun and Lan, 2021; Rampal et al., 2022).**

**Comment 15:** Lines 135: Again are you performing temporal disaggregation? Where for a daily input you are predicting hourly or sub-daily output?

**Author response:** Thanks for your comments. We did not perform time disaggregation. The inputs to the model are daily data and the outputs are also daily data. Equations 1-3 describe that the $i^{th}$ grid in the study area produces p, $\alpha$, and $\beta$ on day $t$. Where $t$ stands for day $t$, $i$ represents the $i^{th}$ grid in the study area. Then $p_i(t)$, $\alpha_i(t)$ and $\beta_i(t)$ are used to obtain the precipitation of the $i^{th}$ grid on day $t$. Specifically, for the input data on day $t$, after going through the fully connected layer of the model, each grid $(0.1^\circ \times 0.1^\circ)$ in the study area produces three values, i.e., $o^1$, $o^2$, and $o^3$. For example, the three output values for the $i^{th}$ grid point on day $t$ are $o_i^1(t)$, $o_i^2(t)$, and $o_i^3(t)$. Next, $p_i(t)$, $\alpha_i(t)$ and $\beta_i(t)$ are calculated through Equations 1-3. Finally, the precipitation of the $i^{th}$ grid point on day $t$ is estimated using a mixed binomial-log-normal distribution with modeled $p_i(t)$, $\alpha_i(t)$ and $\beta_i(t)$ (Baño-Medina et al., 2020, 2022; Sun and Lan, 2021; Rampal et al., 2022). The estimation process of precipitation at day t for other grid points in the study area is consistent with the above. To express our meaning more clearly, we have rewritten this part in Subsection 3.1.2 of the revised manuscript.

$$p_i(t) = o_i^1(t) \tag{1}$$
$$\alpha_i(t) = exp[o_i^2(t)] \tag{2}$$
$$\beta_i(t) = exp[o_i^3(t)] \tag{3}$$

The rewrite in the paper is as follows: **After the FC layer, The output of each grid point $(0.1^\circ \times 0.1^\circ)$ in the study area can be expressed as follows:**

$$p_i(t) = o_i^1(t), \tag{2}$$
$$\alpha_i(t) = exp[o_i^2(t)], \tag{3}$$
$$\beta_i(t) = exp[o_i^3(t)], \tag{4}$$

**where $t$ stands for day $t$, $i$ represents the $i^{th}$ grid in the study area. $o_i^1(t), o_i^2(t)$ and $o_i^3(t)$ represent the three output values of the $i^{th}$ grid point at day $t$, respectively. $p_i(t)$, $\alpha_i(t)$ and $\beta_i(t)$ are calculated through Equations 2-4. The precipitation of the $i^{th}$ grid point on day $t$ is estimated using a mixed binomial-log-normal distribution with modeled $p_i(t)$, $\alpha_i(t)$ and $\beta_i(t)$ (Baño-Medina et al., 2020, 2022; Sun and Lan, 2021; Rampal et**

al., 2022).

**Comment 16:** Equation 6: This could be combined in one expression with equation (2).

**Author response:** Thanks for your nice suggestions. As you said, Equation 2 and Equation 6 in the original paper can be combined. However, Equation 2 is the probability density function of Bernoulli–gamma. We use it here to describe the characteristics of precipitation distribution in detail, and specifically introduce the definitions of parameters p, $\alpha$, and $\beta$. This sets the stage for the later introduction of obtaining the parameters p, $\alpha$, and $\beta$ through the fully connected layer and estimating the precipitation from p, $\alpha$, and $\beta$. Equation 6 defines the loss function used by the model. Here we describe in detail the role of the loss function. We also show that the models used in this paper are run towards minimizing the negative likelihood logarithm of Equation 6 during training. In addition, we also refer to the format and narrative style of Cannon (2008). Therefore, we believe that it may be more consistent with the logic of our paper and the order of introduction to express Equation 2 and Equation 6 separately.

**Comment 17:** Lines 170: Are you using any regularization, as these methods can overfit very easily? Please clarify if not.

**Author response:** Thanks for your comments and suggestions. We use the early stopping strategy to regularize the model to prevent overfitting, where patience is set to 30. This has been clarified in subsection 3.1.4 of the original manuscript. To express our meaning more clearly and streamline the structure of the paper, we deleted subsection 3.1.4 and merged the contents of subsection 3.1.4 into subsection 3.2 in the revised manuscript.

The modification is as follows:

**3.2 Models for comparison and experimental parameter setting**

**We compare the proposed RRDBNet with a generalized linear regression (GLM) method and two deep learning-based methods including CNN (Baño-Medina et al., 2020), and RDBNet (Zhang et al.,2018; Wang et al.,2018).**

**RRDBNet: The network structure is shown in Fig. 2, where the specific parameters of each layer are configured as displayed in Supplementary Table 1.**

**GLM: Generalized linear regression model (GLM) uses binomial family and linked logit**

to predict the occurrence of precipitation and gamma-based family and linked logit to predict the amount of rainfall, respectively. Finally, the occurrence of rainfall is multiplied by the amount of rainfall to obtain the final rainfall forecast. In this paper, the GLM considers the predictors of the four grids that are nearest neighbors to the target location (Baño-Medina et al., 2020).

CNN: The CNN is derived from the model CNN1 proposed by Baño-Medina et al. (2020) for statistical downscaling of precipitation prediction. Baño-Medina et al. (2020) also compared the CNN1 with some variants of the CNN model and found that the CNN1 performs well in the European domain for precipitation downscaling.

RDBNet: The RRDB in the proposed RRDBNet is composed of RDB performing multilevel residual learning. We replace the RRDB with a single RDB, and the network model formed after the replacement is denoted as RDBNet. RDBNet is employed as a comparison model to verify whether the multi-level residual learning strategy of RRDB is effective in precipitation downscaling.

To ensure the fairness of the comparison, we made uniform experimental parameter settings for the deep-learning models. The learning rate set to $10^{-4}$, and the batch size set to 32. The loss function uses the function defined by Eq.5 to minimize the negative log-likelihood of the Bernoulli-gamma distribution. We use the early stopping strategy to regularize the models to prevent overfitting, where patience is set to 30. During the training period, the best model weight parameters are saved. And during testing period, the model calls this best parameter weight to test the data.

**Comment 18:** Lines 180: These metrics are commonly known, so we do not need these in the text. If you'd like to include them add them to the supplementary section or appendix.

**Author response:** Thanks for your comments and suggestions. We have added them to the Supplementary Section and removed subsection 3.3 (**Evaluation matrix**) in the revised manuscript.

**Comment 19:** Lines 185: This is a very small validation sample (4 years), which makes it challenging to examine the performance on extremes. I'd recommend having a supplementary

**Author response:** Thank you for your comments and suggestions. Based on your suggestion, we have did the experiments in which the test period was extended to 10 years (2011-2020), and the training period was 2001-2011. The experimental results are shown in Table R1. It can be seen that RRDBNet outperforms the other models on R95P, R99P and RX1Day with the smallest RMSE and the largest CC. Therefore, RRDBNet has good performance in capturing extreme precipitation. We also performed a corresponding analysis in the Discussion Section of the revised manuscript. The corresponding analysis in the Discussion Section is as follows: **In addition, we also did the experiments in which the test period was extended to 10 years (2011-2020), and the training period was 2001-2011. We further evaluated the performance of the models in extreme precipitation and the results are in Supplementary Table 7. It can be observed that RRDBNet also performs well compared to other models when the test period is extended.**

**Comment 20:** Lines 185: "downscaling projections" – you are not downscaling projections?

**Author response:** Thanks for pointing out the mistake. We have changed "downscaling projections" to "downscaling" in the revised manuscript. The change is as follows: **Fig. 5 shows the spatial distribution of annual mean precipitation for GPM and downscaling based on the four models from 2016 and 2020.**

**Comment 21:** Table 2: I think some measure of uncertainty is required perhaps. I'd suggest repeated the experiment 20 times (with different random seeds) and investigate whether your results are statistically significant. This is importance, as this is the premise of your entire results.

**Author response:** Thanks for your comments and suggestions. Due to limitations of computational resources and time, we only repeated the experiment ten times (with different random seeds). We found that the changes in the results of the ten experiments are very small as in Table R2. Then, we calculated the mean and variance of the ten experimental results and filled them in Table R2 and Table R3. The statistical significance of our results can be demonstrated from Tables R2 and R3.

**Comment 22:** Table 4: I don't think table 4 is useful, this could easily go in the supplementary section.

**Author response:** Thank you for your comments. We have moved Table 4 into the Supplementary Section and labeled it Supplementary Table 2.

**Comment 23:** Figure 6: This seems excessive, I suggest making a (2 x 5) plot and combining this with Figure 5. Where the columns are the models (e.g. GPM, GLM) and the row is the climatology and bias.

**Author response:** Thanks for your suggestions and comments. We consider that if Figure 5 and Figure 6 were combined into a single 2×5 figure, it might make each subfigure very small. So to make each subfigure show more clearly, we merge Figure 5 and Figure 6 as follows:

[Figure]

Figure R2. Spatial distribution of annual mean precipitation (mm/day) in the MRYR from 2016 to 2020 for (a) GPM, (b) GLM, (c) CNN, (d) RDBNet, and (e) RRDBNet. Percentage differences in annual precipitation (%) between model and GPM from 2016 to 2020 in the MRYR for (f) GLM, (g) CNN, (h) RDBNet, and (i) RRDBNet. Percentage Difference=(MODEL-GPM)/GPM × 100%.

**Comment 24:** Figure 5 & 6. It seems a little concerning that there is so much noise in your predictions from the GLM, CNN and other ML models. Other papers do not show such "noise". Are you training your models enough or too much and that your models are overfitting? Some clarification on why there is more noise is needed.

**Author response:** Thanks for your comments and suggestions. Noise appears in Figures 5 and 6. It is because our validation time may be a bit short, only five years (2016-2020). When we increase the time of validation for years (2011-2020), the noise decreases significantly as in Figure R3. We have added the corresponding content in the section Conclusions and discussion.

[Figure]

Figure R3. Spatial distribution of annual mean precipitation (mm/day) in the MRYR for GPM in different testing periods.

**Comment 25:** Figure 7: Plot the bias instead of the raw amounts. Reorder axes from DJF, MAM, JJA, SON.

**Author response:** Thanks for your nice suggestions. We drew Figure 7 just to show that precipitation in the middle reaches of the Yellow River is mainly concentrated in summer and autumn, and then to prepare for the later analysis focusing on the models' performance in extreme precipitation in summer and autumn. In the revised manuscript, we also plotted the bias of the modeled data relative to the observation as shown below in Figure R4. We find that GLM and RRDBNet have a small Difference from observation in summer and autumn precipitation compared to the other models. And the Difference of RRDBNet is minimum in summer.

[Figure]

Figure R4. (a) Seasonal variation of precipitation (mm/day) in the MRYR from 2016 to 2020 for GPM, GLM, CNN, RDBNet, and RRDBNet. (b) Differences in Seasonal precipitation between model and GPM from 2016 to 2020 in the MRYR for GLM, CNN, RDBNet, and RRDBNet.

**Comment 26:** Figure 8: Not useful, this could go in the supplementary instead.

**Author response:** Thanks for your comments and suggestions. We have moved Figure 8 to the Supplementary Section in the revised manuscript and noted it as Supplementary Figure 2.

**Comment 27:** Lines 230: Use either R99P or R95P in the analysis, not both. If you'd like you could keep one in the supplementary section.

**Author response:** Thanks for your great suggestions on improving our manuscript. Based on your suggestions, we have retained only R95P and moved R99P to the Supplementary Section

in the revised manuscript.

**Comment 28:** Figure 9: Frequency at 100mm is very large in Figure 9b and 9d, is there something wrong in your analysis, this is very concerning?

**Author response:** Thank you very much for your comments. Frequency at 100mm is very large in Figure 9b and 9d. This is because we have accumulated the frequency of precipitation above 100mm. The frequency accumulation value is placed at 101mm. Therefore, there will be a big change in the frequency of precipitation above 100mm in Figure 9b and 9d. To express our meaning more clearly, we have made a note in the caption of this figure in the revised manuscript. The modification is shown in Figure R5 below.

[Figure]

Figure R5. (a) Comparison of probability density functions of daily precipitation from 1 mm to 50 mm for all grids of GPM, GLM, CNN, RDBNet, and RRDBNet in the MRYR from 2016 to 2020. (b) The same as (a) but from 50 mm to 100 mm. (c) and (d) The same as (a) and (b) but for observation from 149 stations and nearest grids of GPM and RRDBNet in the MRYR from 2016 to 2019. In (b) and (d), the frequencies of precipitation above 100 mm are cumulated and the cumulative values are placed at 101 mm.

**Comment 29:** Figure 10: Again only one of the R95 and R99 plots should be plotted. You should also compute the bias of the R95 fields against the observations (the percentage bias) in one single plot. It seems strange that you have so much noise in your plots. Figure 11: Again combine with Figure 10.

**Author response:** Thanks for your comments and suggestions. In the revised manuscript, the figures for R95P have been retained in the text and the figures for R99P have been moved to the Supplementary Section. Figure 11 in the original manuscript represents the percentage bias of the R95 fields against the observation. We have merged Figure 10 and 11 together in the revised manuscript. The display is shown in Figure R6 below. There is noise in the picture. It may be because the period of our test was a bit short, only five years (2016-2020). We further conducted verification for 10 years (2011-2020) as shown in Figure R7. It can be seen that the noise is significantly reduced.

[Figure]

Figure R6. Spatial distribution of R95P in (mm) the MRYR from 2016 to 2020 for (a) GPM, (b) GLM, (c) CNN, (d) RDBNet, and (e) RRDBNet. Percentage differences in R95P (%) between model and GPM from 2016 to 2020 in the MRYR for (f) GLM, (g) CNN, (h) RDBNet, and (i) RRDBNet.

[Figure]

Figure R7. Spatial distribution of extreme precipitation in the MRYR for GPM in different testing periods.

**Comment 30:** Table 5: This could be in the supplementary section or combined with information in Figure 10. Table 6: Not required, could be supplementary.

**Author response:** Thank you for your valuable comments. Tables 5 and 6 show the performance of the model in spatial for R95P and R99P. Spatially, RRDBNet reduces RMSE by 58% (79%) and improves CC by 38% (145%) relative to GLM for R95P (R99P). This improvement is very noticeable. RRDBNet also outperforms CNN and RDBNet on R95P and

R99P. It has the smallest RMSE and largest CC. Therefore, we believe that Tables 5 and 6 are important to prove the performance of our model. It might be more appropriate to put them in the main text.

**Comment 31:** Figure 12: Not required, and again very interesting why the outputs are so noisy. Figure 13: Not required, could be supplementary.

**Author response:** Thank you very much for your comments and suggestions. We have combined Figures 12 and 13 into a single figure, placed it in the Supplementary section in the revised manuscript and labeled the Supplementary Figure 3. Regarding the noise in the outputs, we have explained it in detail in Comment 29.

**Comment 32:** Figure 16, should be in the supplementary section.

**Author response:** Thanks for your suggestion. Figure 16 is from subsection 4.4 (**Comparison of convergence**). This subsection compares the performance of the models from another aspect, i.e., the convergence speed of the models. The results are shown in Figure 16. We found that RRDBNet converges faster than CNN during the training period and can reach convergence earlier. The fast convergence speed of the model means that it can reduce training time and save computing resources. This also proves that RRDBNet is an effective downscaling tool. Therefore, we think it may be better to put Figure 16 in the main text.

**Comment 33:** Figure 15: This is not a commonly used validation metric. I would validate against the RX1Day (wettest day of the year).

**Author response:** Thanks for your comments and suggestions. We added the validation of the model downscaling results on the RX1Day metrics to the Discussion Section of the revised manuscript. The results are shown in Figures R8 and Tables R4. It can be seen that in the RX1Day metric, RRDBNet also has a very clear advantage over other models, having the smallest Difference, the smallest RMSE and the largest CC.

[Figure]

Figure R8. Variations of annual and monthly precipitation (mm/day) in the MRYR during 2016-2020 for RX1Day. (a) represents wettest day of the year. (b) represents wettest day of the month.

Table R4. Evaluation metrics of RX1Day for each method from 2016 to 2020

|  | Models | Difference | RMSE | CC |
|---|---|---|---|---|
| | GLM | 16.32 | 17.83 | -0.13 |
| | CNN | 4.90 | 6.08 | 0.69 |
| Annual | RDBNet | 9.83 | 11.30 | 0.25 |
| | RRDBNet | 0.03 | 2.65 | 0.91 |
| | GLM | 1.54 | 5.71 | 0.91 |
| | CNN | 0.73 | 4.68 | 0.94 |
| Monthly | RDBNet | 1.09 | 5.86 | 0.92 |
| | RRDBNet | -0.59 | 3.99 | 0.94 |

**Comment 34:** Figure 14: This should be the bias instead of the average precipitation.

**Author response:** We sincerely appreciate your valuable comments. The purpose of Figure 14 is to illustrate that extreme precipitation in the middle reaches of the Yellow River is also concentrated in summer and autumn. This is consistent with the actual situation in the area. Additionally, based on your suggestions, we also plotted the differences of extreme precipitation relative to observation in the revised manuscript. The display is as in Figure R9. It can be seen that the differences of the deep-learning models are smaller than GLM in summer. In autumn, the difference of RRDBNet is the smallest.

[Figure]

Figure R9. Seasonal variations of extreme precipitation (mm) in the MRYR from 2016 to 2020 for (a) R95P.

**Comment 35:** Table 7 & 8, against too much information, this should be in the supplementary

section.

**Author response:** Thanks for your nice suggestions. We have put Tables 7 and 8 in Supplementary Section in the revised manuscript. Table 7 is labeled as Supplementary Table 3 and Table 8 is labeled as Supplementary Table 4.

References

Baño-Medina, J., Manzanas, R., and Gutiérrez, J. M.: Configuration and intercomparison of deep learning neural models for statistical downscaling, Geosci. Model Dev., 13, 2109–2124, 2020.

Baño-Medina, J., Manzanas, R., Cimadevilla, E., Fernández, J., González-Abad, J., Cofiño, A. S., and Gutiérrez, J. M.: Downscaling multi-model climate projection ensembles with deep learning (DeepESD): contribution to CORDEX EUR-44, Geosci. Model Dev., 15, 6747–6758, 2022.

Beniston, M., Stephenson, D. B., Christensen, O. B., Ferro, C. A., Frei, C., Goyette, S., Halsnaes, K., Holt, T., Jylhä, K., Koffi, B., et al.: Future extreme events in European climate: an exploration of regional climate model projections, Clim. Change, 81, 71–95, 2007.

Booij, M. J.: Extreme daily precipitation in Western Europe with climate change at appropriate spatial scales, Int. J. Climatol., 22, 69–85, 2002.

Cawley, G. C., Janacek, G. J., Haylock, M. R., and Dorling, S. R.: Predictive uncertainty in environmental modelling, Neural Netw., 20, 537–549, 2007.

Cleveland, W. S., Grosse E ., and Shyu W. M.: Local regression models. Statistical models in S. Routledge, 309-376. 2017.

Gutiérrez, J. M., San-Martín, D., Brands, S., Manzanas, R., and Herrera, S.: Reassessing Statistical Downscaling Techniques for Their Robust Application under Climate Change Conditions, J. Clim., 26, 171–188, 2013.

Haylock, M. R., Cawley, G. C., Harpham, C., Wilby, R. L., and Goodess, C. M.: Downscaling heavy precipitation over the United Kingdom: a comparison of dynamical and statistical methods and their future scenarios, Int. J. Climatol., 26, 1397–1415, 2006.

Makungo, R., and Odiyo, J. O.: Application of non-parametric regression in estimating missing

daily rainfall data. International Journal of Hydrology Science and Technology 9.3, 236-250, 2019.

Manzanas, R., Lucero, A., Weisheimer, A., and Gutiérrez, J. M.: Can bias correction and statistical downscaling methods improve the skill of seasonal precipitation forecasts?, Climate dynamics, 50, 1161–1176, 2018.

Pan, B., Hsu, K., AghaKouchak, A., and Sorooshian, S.: Improving Precipitation Estimation Using Convolutional Neural Network, Water Resour. Res., 55, 2301–2321, 2019.

Panjwani, S., Naresh Kumar, S., and Ahuja, L.: Bias Correction of GCM Data Using Quantile Mapping Technique, in: Proc. Int. Conf. Comput. Commun. Technol., ICCCT, pp. 617–621, Springer, 2021.

Prudhomme, C. and Davies, H.: Assessing uncertainties in climate change impact analyses on the river flow regimes in the UK. Part 1: baseline climate, Clim. Change, 93, 177–195, 2009.

Rampal, N., Gibson, P. B., Sood, A., Stuart, S., Fauchereau, N. C., Brandolino, C., Noll, B., and Meyers, T.: High-resolution downscaling with interpretable deep learning: Rainfall extremes over New Zealand, Weather and Climate Extremes, 38, 100 525, 2022.

Rodrigues, E. R., Oliveira, I., Cunha, R., and Netto, M.: DeepDownscale: A deep learning strategy for high-resolution weather forecast, in: Proc. - IEEE Int. Conf. eScience, e-Science, pp. 415–422, IEEE, 2018.

Stojkovic, M., Simonovic, S.P.: Mixed General Extreme Value Distribution for Estimation of Future Precipitation Quantiles Using a Weighted Ensemble - Case Study of the Lim River Basin (Serbia). Water Resour Manage 33, 2885–2906, 2019.

Sun, L. and Lan, Y.: Statistical downscaling of daily temperature and precipitation over China using deep learning neural models: Localization and comparison with other methods, International Journal of Climatology, 41, 1128–1147, 2021.

Wand, M. P., and Jones M. C.: Kernel smoothing. CRC press, 1994.

Williams, P.: Modelling Seasonality and Trends in Daily Rainfall Data, NIPS, 10, 1997.